# Evaluation of the Interaction-Based Hazard Index Formula Using Data on Four Trihalomethanes from U.S. EPA’s Multiple-Purpose Design Study

**DOI:** 10.3390/toxics12050305

**Published:** 2024-04-23

**Authors:** Richard C. Hertzberg, Linda K. Teuschler, Anthony McDonald, Yusupha Mahtarr Sey, Jane Ellen Simmons

**Affiliations:** 1Biomathematics Consulting, Atlanta, GA 30307, USA; hertzberg@comcast.net; 2LK Teuschler & Associates, St. Petersburg, FL 33707, USA; lindateuschler@gmail.com; 3Public Health and Integrated Toxicology Division, Center for Public Health and Environmental Assessment, Office of Research and Development, U.S. Environmental Protection Agency, Research Triangle Park, NC 27711, USA; sey.yusupha@epa.gov (Y.M.S.);

**Keywords:** synergy, antagonism, weight-of-evidence, binary mixture interactions, dose addition, environmental chemicals, mixture risk characterization, risk screening

## Abstract

The interaction-based hazard index (HI_INT_), a mixtures approach to characterizing toxicologic interactions, is demonstrated and evaluated by statistically analyzing data on four regulated trihalomethanes (THMs). These THMs were the subject of a multipurpose toxicology study specifically designed to evaluate the HI_INT_ formula. This HI_INT_ evaluation uses single, binary and quaternary mixture THM data. While this research is considered preliminary, the results provide insights on the application of HI_INT_ when toxicology mixture data are available and on improvements to the method. The results for relative liver weight show the HI_INT_ was generally not conservative but did adjust the additive hazard index (HI) in the correct direction, predicting greater than dose-additivity, as seen in the mixture data. For the liver serum enzyme endpoint alanine aminotransferase, the results were mixed, with some indices giving an estimated effective dose lower than the observed mixture effective dose and others higher; in general, the HI_INT_ adjusted the HI in the correct direction, predicting less than dose-additivity. In addition, a methodological improvement was made in the calculation of maximum interaction magnitude. Suggested refinements to the HI_INT_ included mixture-specific replacements for default parameter values and approaches for supplementing the usual qualitative discussions of uncertainty with numerical descriptions.

## 1. Introduction

Risk assessment of environmental chemical mixtures is not new [1]. In fact, their assessment is specifically required in environmental laws, such as the Toxic Substances Control Act of 1976, the Comprehensive Environmental Response Compensation and Liability Act of 1980, the Food Quality Protection Act of 1996, and the Safe Drinking Water Act Amendments of 1996. Mixture assessment approaches for these laws and their updates continue to be developed. In addition, several laws in Europe require consideration of mixture effects [2] and several European agency guidelines include approaches or specific calculation methods to address mixtures [3,4]. While mixture exposures and joint toxicity can be complex processes, the most often used risk assessment approaches are simple and usually have involved some version of additivity of component doses or responses. Example publications include those on the development and application of component-based approaches. In general, for cancer risk assessment at low exposure levels, the default has been a probabilistic approach with the assumption of toxicological independence of the mixture components, with the recommendation to apply response addition. Typically, for noncancer risk assessment of chemicals causing toxicity in similar target organs or systems, the default assumption is toxicological similarity of the components, and the recommendation is to apply some form of dose addition [5,6,7,8,9]. The focus here is on noncancer endpoints and the hazard index (HI) approach, which is based on dose addition [10,11], particularly on the modified version, termed the interaction-based hazard index (HI_INT_), which incorporates toxicological interactions [5,12,13]. Because the HI_INT_ formula is a modification of the HI formula (each hazard quotient is modified), it was expected to perform best on chemicals that were toxicologically similar so that the modified terms would behave as numerical corrections to dose addition.

In this research, a group of scientists from several organizations, under leadership of the U.S Environmental Protection Agency (U.S. EPA), developed a toxicology study protocol to investigate the ability of three specific component-based formulas to correctly predict mixture response, described here as either the probability of an outcome or the measured toxic effect [14]. The study was designed to evaluate the HI_INT_ approach, various methods for detecting departures from dose-additivity, e.g., [15], and proportional-response addition [16]. In this study, hepatotoxicity was evaluated in CD-1 female mice exposed to four trihalomethanes (THMs) as single chemicals, binary mixtures, and quaternary mixtures. This publication reports the results of evaluating the HI_INT_ formula. 

The HI_INT_ has been available to risk assessors as an alternative method to the HI since its first publication in the U.S. EPA’s 2000 risk assessment guidance document, which functions as a supplement to the 1986 Guidelines for the Health Risk Assessment of Chemical Mixtures [1,5]. Subsequently, it was included in mixture guidance from the U.S. Agency for Toxic Substances and Disease Registry [12]. It has been mentioned in the scientific literature as a useful tool for evaluating the impacts of environmental mixture exposures. Lin et al. [17] used the HI_INT_ to evaluate the interaction effects of arsenic, zinc and copper in metal-contaminated fish in Taiwan coastal areas, and Omrane et al. [18] included use of the HI_INT_ in their occupational study research proposal to evaluate interactions of heavy metals. Marx et al. [19] explored environmental exposures to antibiotic mixtures and used the HI_INT_ to quantify increased risks above those generated using dose-addition. Haddad et al. [20] calculated an HI_INT_ for systemic toxicants using data on tissue doses of the mixture constituents. Based on the work of Haddad, Beliveau, Tardif and Krishnan [20], a health risk assessment of THM mixtures from reclaimed water during toilet flushing in China was conducted using the HI_INT_ for noncancer effects [21]. Ryker and Small [22] applied the HI_INT_ to evaluate cardiovascular and neurological interaction effects from exposures to mixtures of arsenic, cadmium and manganese in drinking water as a method for identifying priorities for drinking water research. Finally, Kumari and Kumar [23] reviewed several component-based approaches for mixture risk assessment and identified the U.S. EPA’s HI_INT_ approach as the most appropriate for predicting joint toxicity of chemical mixtures. They also demonstrated the HI_INT_ applicability and challenges using emerging contaminants as an example.

Risk assessment formulas are often found wanting when compared with biologically based dose–response models, but justified because their ease of use and application utilizing available toxicity and exposure data makes them valuable and informative risk assessment tools. Such formulas are not often evaluated for descriptive or predictive accuracy, usually because of a lack of sufficient data, and the HI_INT_ formula is no exception [13]. Our goal is to provide that evaluation, with the scope here limited to the quantitative behavior of the U.S. EPA [5] version of the HI_INT_ as a dose–response formula for predicting mixture toxicity; concepts and mechanisms of interaction toxicology are not considered in this numerical evaluation. 

The method is investigated and demonstrated using data on single chemicals and binary and quaternary mixtures of four THMs: bromodichloromethane (BDCM), chlorodibromomethane (CDBM), bromoform (CHBr3) and chloroform (CHCl3). These four THMs are currently regulated under the EPA’s Stage 2 Disinfection/Disinfection Byproduct Rule. They are appropriate for the evaluation of dose addition-based approaches, as they have several lines of evidence indicating toxicological similarity, including the liver as a sensitive target organ in common across the four chemicals (other endpoints have been observed as well in toxicology studies, and they have been associated with bladder cancer in epidemiology studies) [24,25]. The two hepatotoxic endpoints used to evaluate the HI_INT_ formula in this research are percent relative liver weight (PcLiv) and the serum enzyme endpoint alanine aminotransferase (ALT). The purpose of this research is to illustrate and evaluate the HI_INT_ methodology and make recommendations for improvements to the approach. The purpose is not to develop a health risk assessment of the THM mixture; thus, it would not be appropriate to use these research results regarding toxicity of the THMs to inform a risk assessment. Because of the complexity of the analysis, Table 1 describes the notation we will use herein to present the methodology and results. 

### 1.1. Hazard Index

Among the methods in the U.S. EPA guidance on health risk assessment of chemical mixtures, the most commonly applied approach for characterizing joint risk from multiple chemicals is the hazard index (HI), typically used for chemicals that are toxicologically similar (at least for those having a common target organ). This index is based on the assumption of dose addition, where the toxicity of the mixture exposure can be represented by the toxicity of the sum of component doses, each scaled by its relative toxic potency. The common implementation of the HI by the U.S. EPA is the formula used for Superfund sites [11], where each component exposure is scaled by the inverse or its toxicity-based reference value, i.e., U.S. EPA’s Reference Dose (RfD, for oral exposure) or Reference Concentration (RfC, for inhalation exposure). For example, with oral exposures to J chemicals, the HI formula is
(1)$$HI=\sum_{j=1}^{J}\frac{E_{j}}{RfD_{j}}$$
where both the exposure level, *E*, and *RfD* are in the same units so that *HI* is dimensionless. One short-hand presentation of this formula uses the hazard quotient (*HQ*) for each component ratio:(2)$$HI=\sum_{j=1}^{J}HQ_{j}$$
where in this oral exposure example *HQ* = *E*/*RfD* for each component. When exposures are such that *HI* = 1 or less, the combination exposure is considered to pose no significant health concern. When *HI* > 1, further investigation could be indicated (see ATSDR [12] for example types of such investigation) or controls could be brought in to lower the exposures until *HI* = 1 or less (see p. 2–12 of U.S. EPA dose addition White Paper [9]). As an example, the European Chemicals Agency (ECHA) proposed a tiered approach for practical risk assessment of biocidal products, which is based on refinement steps for calculated HQs and HIs [26]. As defined by the U.S. EPA, dose addition is expected to occur when the component chemicals act as dilutions or concentrations of each other [5]. That degree of similarity supports use of true dose addition formulas, such as those with relative potency factors that allow numerical estimation of mixture response [9]. The HI assessment, with less evidence of toxicological similarity, is likewise interpreted with less clarity, serving only as a decision index. 

Because the HI is an index of concern, not an estimate of a toxic response or probabilistic risk, the statistical evaluations related to its scientific support usually focus on its basic concept of dose addition and toxicological similarity of the component chemicals. Quantitative evaluations are complicated by the toxicological judgments and science policy decisions embedded in the reference values (e.g., RfD’s use of uncertainty factors and lower confidence limit on ED10). Statistical evaluations of the index with mixture data can be performed and are easiest to interpret when the doses and responses are in the same organism. For example, if the ED10 for the rat is used in the potency weighting instead of the RfD, then it is easy to show that, within the domain of the experimental rat data, *HI* = 1 is equivalent to the Berenbaum [27] definition of dose additivity, and thus, when the mixture component doses give *HI* = 1, the mixture is at its ED10 value for the rat [10,28]. One evaluation of the consistency of a mixture with dose additivity then compares the predicted ED10 calculated from dose addition (the harmonic mean of the component ED10 values weighted by their fractions) with the observed ED10 directly estimated from the mixture data. That comparison is clearer and simpler along a fixed ratio ray, so that the combination of component doses can be replaced by the total mixture dose (sum of unscaled component doses). In the following sections, the HQ is assumed to use the EDx in the denominator. 

The advantages of the HI include its simplicity and its easily met information requirements. Its main disadvantage is its constrained application to toxicologically similar chemicals. Most environmental exposures involve many types of chemicals with possibly different endpoints, so that similarity constraint means several organ-specific HI values may need to be calculated, assuming such organ-specific data are available and if that level of specificity is necessary for decision-making [11]. Furthermore, toxicological interactions are not addressed. With the goal of broader applicability and more realistic risk assessment, the U.S. EPA developed a modification to the HI that incorporates interactions [5]. The purpose of this paper is to evaluate the quantitative ability of the formula for that modified index (*HI*_INT_) to predict the toxicity of a mixture, to suggest improvements to the formula and to identify key uncertainties with the use of *HI*_INT_ for human health risk assessment.

### 1.2. HI_INT_

The U.S. EPA’s 2000 mixtures guidance document puts forth an approach for incorporating toxicologic interactions into the mixture risk assessment by using a modified *HI*, denoted by *HI*_INT_, that reflects the available evidence on pairwise toxicologic interactions [5,29]. The formula for *HI*_INT_ has the *HQ* for each chemical multiplied by an adjustment function that represents all possible pairwise interactions with that chemical (thus the constraint of k not equal to j).
(3)$$HI_{INT}=\sum_{j=1}^{J}HQ_{j}\left(\sum_{k\neq j}^{J}f_{jk}M_{jk}^{B_{jk}g_{jk}}\right)$$

The factors in the adjustment function (second summation) in Equation (3) relate to the likely magnitude and direction of the interactive influence and incorporate the influence of unequal mixing ratios. They are described in detail elsewhere [5,13,29] and will only be summarized here.

*f*_jk_ is the toxic hazard of the kth chemical relative to the total hazard from all chemicals potentially interacting with chemical j.
(4)$$f_{jk}=\frac{HQ_{k}}{\left(\sum_{j=1}^{J}HQ_{j}\right)-HQ_{j}}$$

The function *f* is a normalizing function that ensures the modifying summation is numerically constrained. For example, if all chemical pairs are dose additive so all *M*^Bg^ = 1, then *f* ensures that the inner sum = 1 and so, the *HI*_INT_ in Equation (3) is equal to the *HI* in Equation (1). 

*M*_jk_ in Equation (3) is the theoretical maximum interaction magnitude, estimated from binary interaction data as the ratio of isoeffective doses (e.g., ED10s), comparing the observed EDx from the mixture data with the mixture EDx predicted from components using dose addition, then adjusted using the component fractions (see details in Equations (14) and (15)). It can be asymmetric, i.e., the magnitude of chemical j influencing the toxicity of chemical k can be different from the magnitude of chemical k influencing the toxicity of chemical j. The U.S. EPA default is 5, a value that is consistent with a review of greater-than-additive interactions at low doses [30]. However, the value of *M* for any specific mixture can be calculated whenever appropriate single-chemical and binary mixture data are available. *M* is defined as the larger of the ratio of predicted to observed and observed to predicted, so that *M* is always greater than or equal to one and thus, is the multiplicative or “n-fold” change (vs. dose addition) in mixture EDx caused by the interaction. The direction of interaction (whether observed is greater or less than predicted) is shown in the sign of parameter *B*. The details of how *M* is calculated are presented below. Note that the interaction magnitude is assumed to be constant over the range of mixture doses of interest.

*B*_jk_ is the binary weight of evidence score for the strength of evidence that chemical k will influence the toxicity of chemical j in humans. This subjective approach is used by the U.S. EPA and the U.S. Agency for Toxic Substances Disease Registry [5,10,12]. The value of *B* is negative or positive based on whether the observed binary mixture response is less than or greater than predicted by dose addition, respectively. For the calculations used here, this direction of interaction is determined as the sign of the EDx predicted from components using dose addition minus the observed EDx. The terms “antagonism” and “synergism” are not recommended for interactions because of their potential misinterpretation and vague definition [31]. The range for |*B*| is 0 (no evidence of toxic interaction) to 1 (perfect evidence). The U.S. EPA numerical values for *B* are based on a simple judgment of the extent of extrapolation from the toxicity data to human response (Table 2). For example, it has lower values when using in vitro data or information on a different exposure route from the scenario being addressed. The scores err on the side of safety in that, except for excellent evidence, the influence of less-than-additive interaction is smaller than its counterpart for greater-than-additive interaction. 

*g*_jk_ is the degree to which chemicals j and k are present in equitoxic amounts in the mixture, as judged by their respective *HQ* values. By design, its maximum is when *HQ*_1_ = *HQ*_2_ (Figure 1). The function is the ratio of the geometric mean to arithmetic mean of the *HQ* values.
(5)$$g_{jk}=\frac{\sqrt{HQ_{j}*HQ_{k}}}{\left(HQ_{j}+HQ_{k}\right)/2}$$

The interaction magnitude *M* is the conceptual maximum interaction for the binary mixture, so *g* adjusts (decreases) the magnitude in case the environmental assessment mixture is not of equitoxic composition (thus, *g* uses the fractions in the assessment mixture). This assumption of greatest interaction when component doses are equitoxic is also explicit in Finney’s model of a deviation from dose additivity [32]. (However, we note that this is an untested assumption that needs further toxicity testing and quantitative investigation to confirm.)

These two modifying functions *f* and *g* are each completely determined by the *HQ*s of the component chemicals. The function *g* is symmetric with regard to the two components in that the indices (subscripts) can be in either order, i.e., *g*_jk_ = *g*_kj_. In contrast, the function *f* is not symmetric and neither is the binary interaction magnitude *M*, nor the weight of evidence (WOE) score *B*; all are directional in that each has two values: one refers to the influence of chemical 1 on the toxicity of chemical 2 and the other refers to the influence of chemical 2 on the toxicity of chemical 1. This asymmetry is most easily shown by the binary WOE. Consider the example of the ATSDR Interaction Profile for Arsenic, Cadmium, Chromium, and Lead [33]. That document includes interaction tables for both directions, as shown in this excerpt from the List of Tables (p. xvii): 


*“Table 18. Effect of Lead on Arsenic: Neurological Toxicity for Oral Exposure*


*Table 19. Effect of Arsenic on Lead: Neurological Toxicity for Oral Exposure”*.

For the chemical pair of lead and arsenic, if labeled as chemicals 1 and 2, respectively, *B*_12_ and *M*_12_ refer to the interactive influence of the second chemical on the toxicity of the first, i.e., the influence of arsenic on the toxicity of lead. When the interaction direction is unknown, the assumption is that *B* and *M* are each symmetric with respect to the component chemicals. 

The formula in Equation (3) has been mathematically proven or demonstrated to behave correctly for the following limiting cases [5,13]: 

No interactions. When *M*_jk_ = 1 (for all j,k), no interaction modification is applied and so *HI*_INT_ = *HI*. 

No evidence of interaction. Then *B*_jk_ = 0 for all j,k, no interaction modification is applied and *HI*_INT_ = *HI*. 

Maximum and identical interaction magnitudes. When all *HQ*s are equal and all binary interaction magnitudes are the same, say, *M*_jk_ = 5 (for all j,k), that five-fold change in effective dose carries over to the index, i.e., *HI*_INT_ = 5 × *HI*.

Dominant component. When one component’s HQ is dominant (e.g., HQ_1_ >> HQ_k_ for all k > 1), then the mixture effect should be similar to that of the dominant chemical and the HI_INT_ is numerically close to the dominant chemical’s HQ (e.g., HQ_1_). 

## 2. Methods

### 2.1. Evaluation Approach

The formula in Equation (3) and its functions f and g are examined here in terms of the general goal of the evaluation: whether the *HI*_INT_ formula, when viewed as a dose–response function, is more consistent with the mixture toxicity data than is the *HI* formula. The specific details of their calculations are in Section 2.3. Using toxicity data on four THMs, described below in Section 2.2, the *HI*_INT_ formula is evaluated in three parts. In part one, a simplified scenario is adopted so that the formula calculations and statistical response estimates for the mixtures are more easily performed and interpreted. In part two, function g is evaluated for plausibility using actual binary mixture data for mixtures with differing component fractions. In part three, the formula is explored in terms of the ability of a function that is only based on components and binary interactions to predict the toxic response of a mixture of more than two components, in this case, a quaternary mixture. 

To clarify the focus on the dose–response function in the index formula and not the index itself, the following sections use *y* for the response instead of the index. First, the HQ notation is also replaced using only the component fractions and total mixture dose (the effective doses *D*j in the denominators are fixed parameters).
(6)$$HQ_{j}=\frac{d_{j}}{D_{j}}=d_{MIX}\frac{\pi_{j}}{D_{j}}$$

Here *π*_j_ = component dose fraction. Similarly, the functions *f* and *g* can be represented using only the component fractions and total mixture dose.
(7)$$f_{jk}=\frac{\pi_{k}/D_{k}}{\left(\sum_{i=1}^{J}\pi_{i}/D_{i}\right)-\pi_{j}/D_{j}}$$
(8)$$g_{jk}=\frac{\sqrt{\pi_{j}\pi_{k}/(D_{j}D_{k})}}{\left(\pi_{j}/D_{j}+\pi_{k}/D_{k}\right)/2}$$

The formula for the *HI*_INT_ can then be represented with arguments of the mixture total dose (*d*_MIX_), component fractions and component effective dose values (*D*). As shown in Section 2.3, when the index = 1, the mixture dose is at its EDx, i.e., *d*_MIX_ = *D*_MIX_. Thus, we solve Equation (9) for *d*_MIX_, resulting in the weighted harmonic mean of the component EDx values. That estimate is then compared with the mixture EDx calculated from the mixture data.
(9)$$1=d_{MIX}\sum_{j=1}^{J}\frac{\pi_{j}}{D_{j}}\left(\sum_{k\neq j}^{J}f_{jk}M_{jk}^{B_{jk}g_{jk}}\right)$$

#### 2.1.1. Simplified Scenario

To reduce the obfuscations of the procedural uncertainties when using the index in Equation (3), we start with the best of cases so that the focus is on the quantitative characteristics of the formula as a dose–response function. 

Use of EDx in the denominator. Instead of calculating *HQ* values by the ratio of exposure to some toxicity-based exposure (dose) value that likely uses uncertainty factors to account for extrapolation and other uncertainties and is intended to be health protective (such as the RfD), the exposure is divided by the *EDx* value, i.e., the dose at which a specified response has been attained. 

No scenario scaling or extrapolation. Instead of estimating human exposures, the evaluation stays in the data space of the experiments, i.e., the raw dose and response values from the actual experiments where the same toxic effect is measured in both the component and mixture experiments. The results then apply to the test animal (e.g., mice), not to humans. 

Excellent weight of evidence for interaction. Because of the other constraints, the information on the interaction weight of evidence is categorized as excellent, i.e., there are no extrapolations needed (same species, same duration, same route, same endpoint, same dose range). This avoidance of extrapolations allows the binary WOE score, *B*, to be set at 1 or −1, depending on the direction of interaction, and thus removes the subjective aspects of the WOE scores so the focus is on the quantitative performance of the formula. 

Two toxic endpoints for the same target organ. The toxicological focus here for the THMs is on liver dysfunction. The endpoint measurements are limited to two: relative liver weight and alanine aminotransferase concentrations in serum, evaluated separately. 

This simplified situation allows the comparison of observed to predicted responses based on the data, in this case the *EDx* values for the specified toxic endpoint. Three benchmark response levels (BMRs) of 10%, 20% and 40% increase over the control mean response are used for each toxic endpoint. For that comparison, we again focus on a single mixture ray (fixed component fractions) so that the *EDx* for the mixture is a single value, the total mixture dose. 

#### 2.1.2. Binary Mixture Evaluation

The binary mixture data are evaluated first to determine if the interaction magnitude varies with the mixing ratios (fractions) of the components. If so, then the function *g* is specifically evaluated for its ability to modify the magnitude, *M*, for the different component ratios. 

#### 2.1.3. Quaternary Mixture Evaluation

Several comparisons are conducted for the four-chemical mixture. For each scenario, the evaluation is for a fixed mixture composition (fixed component fractions) so that the dose–response function uses the total mixture dose instead of the vector of component doses. The quaternary studies include two mixture compositions, allowing two separate evaluations. The mixture dose is then calculated such that *HI*_INT_ = 1. That mixture dose is the predicted *EDx* value, i.e., the dose at a BMR level of x% increase over the control mean for the designated toxic endpoint. The “observed” (modeled) mixture *EDx* for that same response level is then compared with that predicted value, as well as with the mixture dose predicted using the standard *HI* of Equation (1). For the interpretation, we describe each predicted mixture *EDx* as “protective” if it is smaller than the observed mixture *EDx*. To understand the influence of these basis interaction estimates, the quaternary analyses are conducted using interaction magnitudes from 1:1 ratio binary mixtures as well as those from environmental ratio binary mixtures. The final comparison determines whether the *HI* and/or *HI*_INT_ prediction values are protective, or not protective enough, by calculating their values for an assumed mixture total dose = 1 mmol/kg/day. 

### 2.2. THM Data Used for the Evaluation

These THM data were generated by the U.S. EPA as part of a multipurpose study designed to allow the evaluation of mixture risk assessment methods for predicting joint toxicity. Four THMs, drinking water disinfection byproducts, were studied in combination using molar ratios based on an ozonation process followed by chloramination (OZ) and on a chlorination (CL) disinfectant process. The data selected for this investigation included eight binary studies and three quaternary studies, a subset of this series of experiments that investigated the hepatotoxicity of the four THMs in CD-1 mice (Table 3). The environmentally relevant ratios used for the binary and quaternary studies were selected based on real-world proportions of the four THMs in finished drinking water. These ratios represented the average seasonal proportions at 35 water treatment facilities [34]. This resulted in dosages composed of a small percentage of CHBr3 relative to the other 3 THMs in all of the mixture dose groups, except for those using a 1:1 mixing ratio. Further details on experimental design can be found in Teuschler et al. (2000) [14] and Teuschler et al. (2024) [16]. 

#### 2.2.1. Animals and Husbandry

For this research, female CD-1 mice were obtained from Charles River Laboratory (Raleigh, NC, USA) at ~60 days of age, and the animals were used in a facility certified by the American Association for the Accreditation of Laboratory Animal Care. Procedures were approved by the Institutional Animal Care and Use Committee (U.S. EPA, RTP, NC, USA). The animal room was maintained at a temperature of 22 ± 2 °C at a relative humidity of 50 ± 10% and on a 12/12-h light/dark cycle (lights on at 06:00 a.m.). Mice were housed in polycarbonate cages with heat-treated pine-shaving bedding. They were allowed to acclimate to the animal facility for a minimum of 3 days before dosing; Prolab^®^ RMH 3000 (Land O″ Lakes, Inc., Minneapolis-St. Paul, MN, USA) feed and tap water were allowed ad libitum throughout the duration of the experiment. 

#### 2.2.2. Chemicals in the Mixtures Used for Evaluation of HI_INT_

The test chemicals for BDCM, CDBM, CHCl3 and CHBr3 were obtained from Sigma Aldrich Chemical Company (St. Louis, MO, USA). The supplier certified their purity as greater than 98%. The four THMs in this study are considered toxicologically and structurally similar. Their log *p* values are similar, ranging from 1.52 to 1.79; thus, all four are readily eliminated from the body (source: chemspider.com, accessed on 21 March 2024). They all have shown dose-related effects on three serum enzymes indicative of hepatic injury: sorbitol dehydrogenase (SDH), alanine aminotransferase (ALT) and aspartate aminotransferase (AST) and on the liver itself. Three THMs are brominated, and some results show brominated THMs to be more toxic than chlorinated THMs. For example, one acute study [35] found that BDCM caused significantly greater levels of serum hepatotoxicity markers than CHCl3 at 48 h post exposure. They proposed that hepatotoxic potency differences between BDCM and CHCl3 “may be due to pharmacokinetic dissimilarities such as greater metabolism of BDCM to reactive metabolites or more extensive partitioning of BDCM into kidneys and fat depots, resulting in prolonged target tissue exposure”. The four THMs evaluated in this study have dose–response data on the same endpoints (e.g., hepatotoxicity) and co-occur in finished drinking water; thus, they were considered appropriate for a component-based mixture evaluation.

#### 2.2.3. Experimental Design

For each experiment, groups of 8–20 female CD-1 mice (Charles River Breeding Laboratory, Raleigh, NC, USA), 65–70 days old at the start of dosing, were administered THMs in an aqueous vehicle by gavage for 14 days (dosing each day between 8 a.m. and noon) and killed on day 15. Dosing solutions were made fresh daily in “gas-tight” vials immediately prior to dosing. Animals were assigned to treatment groups to ensure no statistically significant difference in body weight between treatment groups at the beginning of the experiment. An aqueous vehicle was used (10% Alkamuls EL-620, also known as Emulphor) to avoid the confounding effects of a corn oil vehicle. Gavage volume was held constant at 10 mL/kg to avoid confounding by varying gavage volumes. The dose metric used was mmol/kg/day. Hepatotoxicity was assessed in the morning following the 14th day of dosing by serum indicators and histopathology.

The endpoints evaluated included body weights, liver weights, liver to body weight ratios (relative liver weights, as percent), and three serum enzymes indicative of hepatotoxicity: sorbitol dehydrogenase (SDH), alanine aminotransferase (ALT) and aspartate aminotransferase (AST). Experiments were conducted for each of the six possible binary combinations of these four THMs and two of these binary combinations were repeated, making a total of eight binary experiments. In each binary study, there was a control group as well as component dose–response data on the two THMs involved, so each component had several candidate datasets for the single chemical dose–response modeling. Finally, three experiments devoted to the four-THM mixture were selected for this study, with compositions reflective of THMs in real-world CL- and OZ-treated waters. For each of the binary mixture experiments in the study, eleven animals were in each dose group; not all animals tested survived.

The experimental design details are shown in Table 3, including the doses and mixing ratios of the THMs in the binary and quaternary mixtures. Binary combinations, identified as experiments 2-1 through 2-6, 2-1A and 2-4A, were composed of 12 dose groups: a vehicle control; three dose levels (0.1, 1.0 and 3.0 mmol/kg/day) of each THM alone; three dose levels (0.1, 1.0 and 3.0 mmol/kg/day) of the binary mixture in a 1:1 mixing ratio; and two dose levels (1.0 and 3.0 mmol/kg/day) of the binary combination at a mixing ratio based on environmentally relevant concentrations. The three quaternary experiments, identified as #4-2, 4-3 and 4-4, each compared a CL mixing ratio to an OZ mixing ratio in order to represent those two common drinking water treatment systems. These ratios were used because their data were available from a “split-stream” drinking water treatment plant in which the same source water underwent the two different disinfection processes. Each of these three experiments contained one CL mixture and one OZ mixture, with each mixture studied at only one dose: 0.05, 1.5 or 3.0 mmol/kg/day for studies 4-4, 4-3 and 4-2, respectively. A one-way analysis of variance showed no statistical difference across the control mean values, so those three datasets were combined. The resulting merged quaternary data for dose–response modeling then included three nonzero doses (Table 3). Each of these three studies also included a control group and a single-dose group for each of the four components tested alone at 0.76 mmol/kg/day for CDBM and 1.5 mmol/kg/day for the other THMs for comparison with the single-THM dose–response curves. Unfortunately, preliminary analysis found that those single-dose results did not compare well with the single-chemical dose–response curves for the same component in the other studies; therefore, the quaternary experiment data are used solely to illustrate the HI_INT_ methodology but are not suitable for drawing conclusions about THM quaternary mixture toxicity.

### 2.3. Calculation Approach

This interaction HI formula uses dose–response values derived from information on the components, the binary mixtures and the binary interaction magnitudes. When data are available, comparisons can then be made with dose–response information on the whole mixture. Figure 2 details the progression of the analysis conducted herein, including specifics for the THM data that are used to evaluate the method. The steps undertaken in the analysis are as follows, using abbreviations as shown above:(1)Conduct data quality analyses, perform dose–response modeling and select the best fitting model for each single, binary and quaternary THM dataset.(2)Obtain, for each binary mixture, the two components’ model estimates for their response-specific doses, i.e., the EDx (*D*_j_) values for the endpoint of interest. Those component *D*_j_ values are estimated using the component-only data from the binary mixture study.(3)Use the component *D*_j_ values to estimate *D*_ADD2_, which is the *EDx[predicted]* under dose addition using the harmonic mean formula (Section 2.3.3).(4)Obtain the observed *D*_MIX2_ estimates for the endpoint of interest by modeling the binary mixture data.(5)Calculate the binary interaction magnitude, *M*_OBS_. Because interaction is defined here as departure of the observed outcomes from the predicted outcomes under dose additivity, its magnitude is found by comparing *D*_MIX2_, the mixture *EDx[observed]*, with *D*_ADD2_, the mixture *EDx[predicted]* that is calculated from dose addition. Calculate the theoretical maximum interaction magnitude [*M* = (*M*_OBS_)^1/*g*^], where the *g* function is based on the component fractions and EDx values in the binary mixture.(6)Calculate and compare *D*_ADD4_ and *D*_INT4_, the predicted mixture EDx values for the two quaternary mixture rays (CL and OZ mixtures), by setting *HI* = 1 and *HI*_INT_ = 1, respectively.(7)Obtain *D*_MIX4_, the observed EDx estimates for the two quaternary mixture rays by modeling the data for each ray. Compare these with the *D*_ADD4_ and *D*_INT4_ values for the two mixture rays.(8)To improve understanding of the potential for this evaluation, present some results in the context of the hazard index. Using exposure information from the quaternary mixture, calculate an *HQ* for each component, endpoint and mean *D*_j_ across the 8 studies. Use these *HQ* estimates to calculate the *HI* and the *HI*_INT_. (Note that the WOE score, *B*, is designated to be 1 or −1, as described earlier. Also note that to calculate *HI*_INT_ for the quaternary mixture, the *g*_ij_ function is based on the component fractions of components i and j in that quaternary mixture.) Compare the *HI* and *HI*_INT_ estimates for an assumed mixture dose = 1 mmol/kg/day. The results from this step show several significant digits to facilitate comparison across indices. In the final conclusions of an actual risk characterization, the *HI* and *HI*_INT_ should be shown rounded to one significant digit.(9)Describe key uncertainties found during the analysis.

#### 2.3.1. Component EDx Values

Each component dataset was fitted by three models: quadratic, 3-parameter exponential, 4-parameter Hill (logistic) model. Model selection was based on judgment of both the lack of fit and the small-sample corrected Akaike information criterion (AICc). When multiple models had acceptable lack-of-fit *p*-values, the difference-scaled AICc was used (*scaled AICc*i = *AICc*_i_ − *AICc*_MIN_) to select the preferred model, with *scaled AICc* < 2 indicating models with the same level of performance [36]. Model fitting was by iterative nonlinear regression, where all datasets were sorted in the same way to reduce the influence of the starting values on the iteration algorithm. The EDx was evaluated using inverse prediction, where the standard error was based on the first-order Taylor series approximation using the inverted dose–response function (JMP Pro v14.2). 

#### 2.3.2. Binary Mixture EDx Values

The same three candidate models used for the components were fitted to each binary mixture using data on a single-mixture ray so that the mixture EDx, as total dose, was a single-dose value (i.e., a scalar, not a vector). The same procedure as with the components was used to select the preferred dose–response model. This contrasts with previous evaluations of dose addition in mixtures where only the preferred component model was applied to the mixture data [37].

#### 2.3.3. Binary Interaction Magnitude

Interaction magnitude, as stated above, is defined here as the U.S. EPA defines it, as the fraction given by the “observed” response-specific dose (e.g., ED10) of the mixture divided by the “predicted” iso-effective dose calculated from dose addition [5]. When “observed” is higher, the fraction is
(10)$$M_{OBS}=\frac{EDx[observed]}{EDx[predicted]}$$
where the observed value is from inverse prediction using the binary mixture model. When “observed” is lower, then the inverse fraction is used, i.e., predicted/observed [5]. This provides a consistent interpretation of interaction magnitude as the “m-fold” change in effective dose, regardless of the direction of the change. 

To reduce the influence of inter-experimental variation when predicting the binary mixture response, all the single-chemical and mixture data came from the same binary experiment. For example, the dose-additive prediction of the response for the CHCl3:CDBM mixture only used the CHCl3 dose–response data that came from the same study that evaluated the CHCl3:CDBM mixture. Because the binary component data and the binary mixture data used to calculate M are from the same study, they include the same control group. Two binary mixtures had replicate studies (CHCl3:BDCM and BDCM:CDBM). In those cases, for each component, the EDx was the average across the replicate studies. For each of these binary mixtures, those two average values were then used to calculate a single predicted mixture EDx value, denoted *D*_ADD_, which applied to both replicate mixtures. For the replicate binary mixtures, the “observed” mixture EDx, denoted *D*_MIX2_, is the average of the two mixture estimates. Using these average values results in the same calculated M for the two replicates. 

In this evaluation, we are using EDx values (say, ED10) in the denominators in Equation (1), and so the HI formula is actually a representation of dose addition. As stated above, when *HI* = 1, the mixture response will be the same response level of the denominators in the formula, i.e., the mixture is at its ED10 [10,28]. With the same simplified notation as above, the EDx value for chemical 1 is denoted *D*_1_. The binary mixture EDx under dose additivity, *D*_ADD2_, will be calculated along a specific mixture ray, the 1:1 composition that we are using. Thus, the component doses, *d*_1_ and *d*_2_, are equal and so the condition of *HI* = 1 can be written
(11)$$\frac{d_{1}}{D_{1}}+\frac{d_{2}}{D_{2}}=d_{1}*\left(\frac{1}{D_{1}}+\frac{1}{D_{2}}\right)=1$$

Along this fixed ray, the mixture dose is the sum of the component doses. For the 1:1 mixture, the mixture dose is 2 × *d*_1_. The calculated mixture EDx under dose addition, *D*_ADD_, is then found by solving Equation (11) for *d*_1_ and multiplying by 2:(12)$$D_{ADD}=2*d_{1}=\frac{2}{\left(\frac{1}{D_{1}}+\frac{1}{D_{2}}\right)}$$

The generalization of Equation (12) for a binary mixture with different component fractions, such as the environmental mixture compositions shown in Table 3, is
(13)$$D_{ADD}=\frac{1}{\left(\frac{\pi_{1}}{D_{1}}+\frac{\pi_{2}}{D_{2}}\right)}$$
where *π* is the component fraction of the mixture total dose, e.g., *π*_1_ = *d*_1_/(*d*_1_ + *d*_2_). That equation is the weighted harmonic mean formula, previously shown for calculating LD50s under dose addition [32]. The estimate of *M*, the binary mixture interaction magnitude, is then as defined above in Equation (10): the fraction given by *D*_ADD2_ calculated from Equation (13) divided by the modeled *D*_MIX2_ estimated from the binary mixture data (or the inverse, whichever is larger). 

There is one more consideration in the calculation of *M*_jk_. The binary interaction magnitude is assumed to have a maximum for equitoxic component exposures, such as the 1:1 mixture used in the presentation above for components with similar toxic potencies. There is minimal empirical support and no biological theory supporting that assumption, so we have termed that value the “conceptual maximum”. Because the U.S. EPA mixture risk guidance does not provide any details for how to calculate *M* when the mixture data are not for equitoxic component doses, we have proposed a method herein and have illustrated it using the THM data. When the interaction magnitude is estimated from binary mixture data where the *HQ* values are unequal, that observed interaction magnitude (according to the above assumption) is less than the conceptual maximum, and so the observed magnitude should be increased to that theoretical maximum before its use in the *HI*_INT_ formula. The assumption for the observed interaction magnitude (*M*_OBS_) is that it has the same relationship to the theoretical maximum as that in Equation (3):(14)$${M_{OBS}}_{jk}={\left(M_{jk}\right)}^{g_{jk}}$$

By raising both sides of Equation (14) to the power 1/*g*, where *g* is based on the component fractions in the binary mixture, we have the predicted maximum magnitude *M* calculated from the observed magnitude. For chemicals 1 and 2:(15)$$M_{12}={\left(M_{OBS,12}\right)}^{1/g_{12}}$$

For any binary interaction magnitude that is estimated from a non-equitoxic mixture, the “observed” value is then adjusted upward to give the theoretical maximum, as shown by Equation (15). When applying the *HI*_INT_ formula in an environmental assessment, any pairs of environmental mixture component *HQ* values that are not equal would then be used in the function *g* to reduce *M* to the expected interaction magnitude for that actual environmental mixture exposure. 

#### 2.3.4. Quaternary Evaluation of EDx and Interaction Magnitude

For the four-chemical mixture considered in this paper, there are six possible binary combinations. These *M* values are then determined for each of the six binary mixtures. Consider the simplified case where no extrapolation is needed and where all binary interactions are greater than dose-additive; thus, *B*_jk_ = 1 for all j,k. The interaction HI for the four-chemical mixture (J = 4 using Equation (3)) is then:(16)$$HI_{INT}=\sum_{j=1}^{4}HQ_{j}\left(\sum_{k\neq j}^{4}f_{jk}M_{jk}^{g_{jk}}\right)$$

All the values in Equation (16) are now available: functions *f* and *g* require only the *HQ* values, which are the fractions *d*_j_/*D*_j_ (component dose divided by the component EDx) for each component. 

The index in Equation (16) is then evaluated by comparing its predicted mixture EDx with the observed EDx that is estimated directly from the model of the quaternary mixture data. There are two such predictions. The first prediction, the dose-additive case, is the same dose-additive evaluation described above for the binary mixtures, but using a four-chemical expansion of Equation (13). For the second prediction, the interactions case, we derive the predicted EDx as the set of component dose levels where *HI*_INT_ = 1. The observed EDx is the estimated EDx from modeling the data on the quaternary mixture. As with the binary mixtures, a simplified evaluation is performed for a single mixture ray (constant component proportions), so that the mixture EDx is a single total mixture dose value, the sum of the four component doses. In a derivation similar to that for Equation (13), by replacing all component doses on the mixture ray with the component fraction times the total mixture dose, the two functions *f* and *g* in Equation (16) become constants. The magnitudes *M* are also assumed constant along the ray. Let the right-hand or inner summation be denoted *W*_j_.
(17)$$W_{j}=\left(\sum_{k\neq j}^{J}f_{jk}{M_{jk}}^{B_{jk}g_{jk}}\right)$$

Then *W*_j_ itself is a constant, and *H*I_INT_ is simply represented by:(18)$$HI_{INT}=\sum_{j=1}^{J}\left(HQ_{j}\times W_{j}\right)$$

The interaction-based estimate of the mixture’s EDx, denoted *D*_INT_, is then found by setting Equation (18) equal to 1 and solving for the mixture dose. Simplifying gives the formula’s prediction:(19)$$D_{INT}=\frac{1}{\sum_{j=1}^{J}\left(\frac{\pi_{j}}{D_{j}}\times W_{j}\right)}$$

## 3. Results

### 3.1. Application to the THM Data

The endpoints selected for evaluation of the *HI*_INT_ formula were percent relative liver weight (PcLiv) and the logarithm of ALT [Log(ALT)], where the log transformation was used to help stabilize the increases in variance with dose found in the ALT dataset. Based on the O’Brien test for homogeneity of variance, the log transformation of ALT was an improvement for most of the component and mixture datasets (from 20/33 to 27/33 showing homogeneity of variance). These two endpoints were chosen because of their toxicological significance as evidence of hepatic effects and because the dose–response data were generally well-behaved regarding homogeneity of variance, outliers and curve shape, making them good candidates for estimating EDx values. There were sufficient data in all of the selected studies to obtain model estimates of EDx for three benchmark responses (BMRs): 10%, 20% and 40% increases over the control mean of PcLiv and ALT, with one exception: no CDBM ED40 estimates could be made for study #2–4 because only one animal survived at the high dose level (3.0 mmol/kg/day), precluding reliable EDx estimates at high response levels. The model results were adequate for evaluation of the change in interaction magnitude with change in response level. 

#### 3.1.1. *D*_j_ for the Components

The fitting of the three candidate models to the component data (Log(ALT) and PcLiv) in each binary study consistently showed the three-parameter exponential model, Exp-3P, to perform well, if not best, for most datasets according to the decision criteria used. When it ranked second, its *AICc* was not significantly different from the slightly better quadratic model (Figure 3). The four-parameter Hill model did not perform well, as there were too few dose groups to adequately define the sigmoid shape. 

The modeled component *D*_j_ estimates for PcLiv and Log(ALT) are shown in Table 4 for BMRs of 10, 20 and 40%, and these estimates, along with their 95% confidence limits are shown in Figure 4 parts a-d. Note that, because of the replicate studies, the single-THM effective dose estimates shown in these figures represent different numbers of studies, i.e., five, four, four and three estimates for BDCM, CDBM, CHCl3 and CHBr3, respectively. In the figures, all of the *D*_j_ estimates for each THM show consistency across studies except for the PcLiv *D*_j_ values for CDBM in Figure 4b (experiment 2-4A), whose confidence limits do not overlap with those from the other three studies at any of the three BMR levels. Table 4 also shows the averages of the replicate studies, 2-1 with 2-1A and 2-4 with 2-4A. One unexpected advantage was that taking the average of the *D*_j_ values for CDBM reduced the influence on the analysis results of the noticeably higher *D*_j_ values seen in Figure 4b. All of the replicate study averages were used, along with the other single-THM *D*_j_ estimates, in the calculation of the predicted *D*_ADD2_ values under dose addition, which were in turn used for estimating the binary mixture interaction magnitudes. The component *D*_j_ estimates often had wide confidence intervals, perhaps due to the small sample sizes. That uncertainty is not incorporated in the calculations of the predicted *D*_ADD2_ estimates for the mixtures. Finally, Table 4 also shows the mean *D*_j_ values for each THM and endpoint across the eight studies, which were used in the calculation of *HQ* values that were then input to the *HI* and *HI*_INT_ calculations. 

#### 3.1.2. *D*_MIX2_ for the Binary Mixtures

As was found for the component data, the fitting of the three candidate models to the binary mixture data consistently showed the three-parameter exponential model, Exp-3P, to perform well, if not best, for most datasets. The modeled *D*_MIX2_ estimates for PcLiv and Log(ALT) at the three BMR levels are shown in Table 5(a,b) for the 1:1 and environmental mixtures, respectively. These estimates, along with their 95% confidence limits, show consistency across studies, as seen in Figure 5a,b. Table 5(a,b) also show the averages of the replicate studies, which are used, along with the other binary THM *D*_MIX2_ estimates, as the observed *D*_MIX2_ values in the calculation of interaction magnitude. For all six binary mixtures, the environmental composition only included two nonzero dose levels. That limited the evidence of curvature and restricted the candidate dose–response models. The resulting EDx estimates for those environmental compositions then have important uncertainty. 

#### 3.1.3. Binary Interaction Magnitudes and Interaction Direction

The binary interaction magnitude is assumed and reflected in the *HI*_INT_ formula of Equation (3) to have a maximum for equitoxic component exposures. Because there is no empirical evidence or biological theory supporting that assumption, we have termed that value the “theoretical maximum”. When component exposures in the assessment mixture are not approximately equitoxic, then that maximum is reduced using function *g* in the *HI*_INT_ formula. Each of these binary THM studies included one mixture composition with a 1:1 mixing ratio (equal component doses) and one with environmental mixing ratios, as summarized in Table 3. Converting those ratios into mixture fractions shows the difference in compositions. For example, in study 2-2 both components in the “equimolar” composition have fractions of 0.5, while CHCl3 dominates the environmental mixture with a fraction of 0.985. The components also vary somewhat in potency, as shown by their *D*10 values in Table 4, so that the 1:1 mixtures are not of equitoxic composition. 

The interaction magnitude calculations and interpretations for PcLiv are demonstrated in Table 6, based on *D*_j_, *D*_ADD2_ and *D*_MIX2_ values for BMR of 10% from the 1:1 binary mixture study. For each chemical pair, we calculated the predicted *D*_ADD2_ under dose addition when the *HI* = 1 per Equation (12), using the single-THM (including replicate average) *D*_j_ values (Table 4). The binary interaction magnitude for THMs j and k was then calculated per Equation (10) by comparing the observed *D*_MIX2_ values from the binary study (Table 5(a)) with the predicted *D*_ADD2_ values. For this research, smaller values of *M* (i.e., <1.10) are judged to be consistent with dose addition, and a value of exactly 1.0 indicates high confidence in a dose-additive joint effect. To determine the direction of the interaction, the observed value, *D*_MIX2_, is subtracted from the predicted value, *D*_ADD2_. The interpretation of direction is that if the observed *D*_MIX2_ is larger than the predicted *D*_ADD2_ (yielding a negative number), then the mixture’s joint toxicity is less than predicted by dose addition, so the interaction is less than dose-additive (and vice versa for greater than predicted joint toxicity). It may be noted that the actual differences between the *D*_ADD2_ and *D*_MIX2_ values shown in Table 6 are small and may not be toxicologically significant, but are used here nonetheless to demonstrate the method.

All comparisons of the interaction magnitudes and directions for PcLiv and Log(ALT) are shown in Table 7 and Table 8, respectively. The *M*_jk_ values shown are the theoretical maximum interaction magnitudes calculated per Equation (15) as *M*_jk_ = *M*_OBS,jk_^1/g^ using the proportions of the binary mixtures to determine the value of *g*. As stated earlier, g_jk_ is the degree to which chemicals j and k are present in equitoxic amounts in the mixture, as judged by their respective *HQ* values. By design, its maximum is when *HQ*_1_ = *HQ*_2_ (Figure 1). Although one might conclude that the 1:1 composition binary mixture *M*_jk_ values should already be at or close to their theoretical maximums, we found that this was not consistently the case. For example, for a total dose of 1.0 mmol/kg/day, CHCl3 and CHBr3 are each present at 0.5 mmol/kg/day, which is the numerator for both of their *HQ* values. The denominator, then, is represented by their *D*_j_ values and is the deciding factor for determining how closely the value of *HQ*_1_ is to *HQ*_2_.

For PcLiv, as shown in Table 4 for binary experiment 2-2, the *D*_j_10s for CHCl3 and CHBr3 are 0.73 and 0.29, leading to *HQ*s of 0.5/0.73 = 0.68 and 0.5/0.29 = 1.72, respectively; thus, both the *D*_j_10 values and their associated *HQ*s appear to be different. Although many of the other *HQ*s for the 1:1 binary combinations do seem close in value, we decided to calculate *M*_jk_ consistently by using the adjustment for *M*_OBS_ for these mixtures. Since the environmental binary mixtures have both different numerators as well as different denominators for their component *HQ*s, it was important that this adjustment be made. All the values presented for *M*_jk_ in Table 6, Table 7 and Table 8 then use these adjusted magnitudes. Similarly, all derived values (e.g., *HI*_INT_, *D*_INT_ in Table 9 and Table 10) also use these adjusted theoretical maximum magnitudes. In the example in Table 11, the values presented for *M*_jk_ use these adjusted magnitudes, with unadjusted values shown as *M*_OBS_.

With only a few exceptions, the *M*_jk_ values for the binary combinations are consistent across the different response levels, especially for the 1:1 binary mixtures (left side of Figure 6). The largest differences of note are in experiment #2-2 for CHCl3 and CHBr3, where the PcLiv *M*_jk_ values are 4.94, 3.24 and 1.00 and the Log(ALT) *M*_jk_ values are 1.36, 2.15 and 2.65 for *D*10, *D*20 and *D*40 levels, respectively. The interaction direction seems to depend on the endpoint. Overall, in Table 7, the directions of interactions for PcLiv are highly varied across studies, revealing no particular pattern. However, for the Log(ALT) analysis, all directions together reveal a pattern of either additive or <additive; the few exceptions of >additive are seen only in experiments 2-4/2-4A (1:1 mixture) and 2-3 and 2-6 (environmental mixtures). The results within specific datasets are also consistent in direction across response level. For 13 of the 24 studies in Table 7 and Table 8, the interaction direction is the same across all three BMR levels. For example, for PcLiv in 1:1 mixtures, we see additive for experiment #2-1/2-1A, <additive for experiment #2-2 and >additive for experiments 2-3 and 2-4/2-4A across all BMR levels, and for Log(ALT) in environmental mixtures, we see additive for experiment #2-1/2-1A, <additive for experiments #2-2, 2-4/2-4A and 2-5, and >additive for experiment #2-3 across all BMR levels. In addition, for seven of the twenty-four studies in Table 7 and Table 8, the direction is either additive or in the same direction across all three BMR levels. In only four studies are the results actually contradictory, showing both <additive and >additive results at different response levels within the same study.

#### 3.1.4. Quaternary Mixture Predicted Indices and Predicted Effective Doses

Predictions for the four-chemical CL and OZ mixtures were based on merging the data for experiments #4-2, 4-3 and 4-4 (Table 3). One-way ANOVA showed no significant differences across the three means for either endpoint and was used to justify the merge. We note that for the purpose of analyzing the HI_INT_ formula, with its assumption of maximal interaction at equitoxic doses, it would have been desirable to have a mixing ratio more closely representing component fractions of 0.25, 0.25, 0.25, 0.25. Instead, the three mixtures represented in the quaternary experiments had a relatively small fraction (<0.05) of CHBr3 (Table 3). Similar to the component and binary mixture data, the fitting of the three candidate models to the merged quaternary mixture data consistently showed the three-parameter exponential model, Exp-3P, to perform well, if not best, for all datasets, so it was used for the modeling. 

Table 9 shows the mixture effective doses for each quaternary mixture, endpoint and BMR level combination. The observed effective dose, *D*_MIX4_, is from the dose–response modeling for PcLiv and Log(ALT) using inverse prediction at each endpoint’s BMR value. For the other calculations in Table 9, the predicted mixture effective dose is calculated by setting *HI* = 1 or *HI*_INT_ = 1, i.e., using Equation (13) or Equation (19), respectively. The binary interaction magnitude values, *M*, vary based on which binary mixture is represented (see Table 7 and Table 8). Consequently, there are two predicted effective dose calculations shown for *HI*_INT_ = 1, one based on the binary 1:1 mixtures, *D*_MIX4_ (1:1), and one based on the binary environmental mixtures, *D*_MIX4_ (Env). For these calculations, the observed interaction magnitude for each binary mixture, *M*_OBS_, is adjusted upward to obtain the estimated maximum magnitude, *M*_jk_, using Equation (15); it is the latter adjusted value that is used with the quaternary component fractions in Equation (19) to obtain the *D*_INT4_ (1:1) and *D*_INT4_ (Env) estimates in Table 9. Those adjusted *M*_jk_ values are similarly used for the *HI*_INT_ (1:1) and *HI*_INT_ (Env) calculations shown in Table 10.

The interpretation of results varies across endpoint and quaternary mixture as follows:

For PcLiv with both mixture compositions, the order of the comparisons in all but one case is
*D*_MIX4_ < [*D*_INT4_ (1:1) or *D*_INT4_ (Env)] < *D*_ADD4_, 
an order that suggests that the *HI*_INT_ and the *HI* are not conservative estimates of risk for these mixtures, but that the HI_INT_ did adjust the HI in the correct direction (i.e., reflecting the >additive binary mixture results). 

For Log(ALT) with the OZ mixture, the comparison order is
*D*_ADD4_ < *D*_INT4_ (1:1) < *D*_INT4_ (Env) < *D*_MIX4_, 
an order that does show conservatism in the *HI*_INT_ and *HI* values, with *HI*_INT_ correctly adjusting the *HI* to reflect the less-than-additive binary mixture results.

For Log(ALT) with the CL mixture, the comparison for the 10% and 20% response levels is
*D*_MIX4_ < *D*_ADD4_ < *D*_INT4_ (1:1) < *D*_INT4_ (Env), 
an order that does not show conservatism in the *HI*_INT_ or *HI* values, but where *HI*_INT_ does correctly adjust the *HI* to reflect the less-than-additive binary mixture results. The comparison for the 40% response level is
*D*_ADD4_ < *D*_INT4_ (1:1) < *D*_MIX4_ < *D*_INT4_ (Env), 
showing conservatism in the *HI*_INT_ (1:1) and *HI* values (but not the *HI*_INT_ (Env) value) and correctly adjusting the *HI* to reflect the less-than-additive binary mixture results.

For the *HI* calculations in Table 10, the quaternary mixture total dose is kept at 1.0 mmol/kg/day and each quaternary mixture composition (CL vs. OZ) is evaluated separately. Each component’s exposure level is then its fraction times the total mixture dose, which then is used in the numerator of its *HQ*. Other total doses are not evaluated here because the results are proportional to the results for a 1.0 total dose. The denominator of each component’s *HQ* is its effective dose, *D*_J_, which depends on which BMR level is chosen (10, 20 or 40%), thus, the three columns in Table 10 for each endpoint and each mixture. With those *HQ* values, we calculated the *HI* value from Equation (2) and the *HI*_INT_ (1:1 mixture) and *HI*_INT_ (environmental mixture) values from Equation (16), with results shown for each quaternary mixture, endpoint and BMR combination. Note that the *HI*_INT_ values were calculated using the theoretical maximum *M*_jk_ values from Table 7 and Table 8, and values of *g* determined using the proportions of the quaternary mixtures. All results were greater than 1.0 except those based on the ED40s for PcLiv, which were ~0.5 for both CL and OZ mixtures for all three hazard indices. For both the CL and OZ mixtures for Log(ALT), the *HI* was always larger than the *HI*_INT_ values, consistent with the general <additive direction of the interaction magnitudes in Table 8, where eight of the ten largest *M*_jk_ corresponded to directions <additive. For the PcLiv endpoint, except for the essentially equal index results using ED20 values, the *HI* was always smaller than the *HI*_INT_ values, consistent with Table 7, where nine of the ten largest *M*_jk_ corresponded to directions >additive.

Differences can be seen when comparing the 1:1 and environmental binary mixture results, as shown in Table 9 and Table 10. For the CL mixture, *D*_INT4_ (Env) values are all larger or, in one case, essentially equivalent to the *D*_INT4_ (1:1) values (Table 9) and, correspondingly, *HI*_INT_ (Env) values are all smaller or equivalent to the *HI*_INT_ (1:1) values for both endpoints (Table 10). For the OZ mixture, the results for the two mixtures differ by endpoint. *D*_INT4_ (Env) values are smaller for PcLiv and larger for Log(ALT) compared to the *D*_INT4_ (1:1) values, the reverse being true for the *HI*_INT_ values. 

#### 3.1.5. *HI*_INT_ Method and Results of Estimating the Theoretical *M* by Adjusting *M*_OBS_

As discussed above, because the observed interaction magnitudes for the environmental binary mixtures are based on component proportions that are assumed to lessen the interaction magnitude, we made an adjustment to each *M*_OBS_ using Equation (15) to estimate the theoretical maximum binary magnitude *M*_jk_ for use in the assessment formula for *HI*_INT_ (see Figure 2). For the 1:1 binary mixtures, the results were virtually identical to those without the adjustment, but there were some differences for the environmental binary mixtures. The results using the environmental binary mixture data are demonstrated in Table 11 for the Log(ALT) endpoint using 10% effective dose values for the CL quaternary mixture. This scenario was selected because it produced the largest *HI* and *HI*_INT_ values in Table 10. For each chemical pair of THMs j and k, an *M*_OBSjk_ value is calculated using the m-fold change from predicted *D*_ADD2_ to observed *D*_MIX2_, per Equation (10). Then the theoretical maximum *M*_jk_ value is calculated per Equation (15) using a *g*_jk_ value per Equation (5) based on *HQ*s for that environmental binary mixture. The following results show three decimal places (and, thus, two to five significant digits) to clarify the numerical calculations for easy replication. In the final conclusions of an actual risk characterization, the *HI* and *HI*_INT_ should be shown rounded to one significant digit. For this mixture, it can be seen that the *HI*_INT_ of 4.44 based on the maximum *M*_jk_ values is slightly smaller than the *HI*_INT_ of 4.921 based on the unadjusted *M*_OBS_ values, and both are lower than the *HI* results of 6.033. In addition, the predicted 10% effective dose values for the three indices of *D*_ADD4_ = 0.166, *D*_INT4_ = 0.203 for *M*_OBS_ and *D*_INT4_ = 0.225 for *M*_jk_ are all higher than the modeled *D*_MIX4_ value of 0.11 mmol/kg/day for this scenario (shown in Table 9). Even though we selected the scenario most likely to show a difference between the theoretical maximum *M*_jk_ and the observed *M*_OBS_ interaction magnitudes, the values produced were fairly similar. The third comparison in Table 11 uses the U.S. EPA default of *M*_jk_ = 5 for each binary mixture (see Section 1.2). The values of *HI*_INT_ of 12.091 and *D*_INT4_ = 0.083 clearly show the conservative nature of that default assumption, but only two of six binary interactions were >add, so the use of the default magnitude fails to reflect the generally less-than-additive results, i.e., *HI*_INT_ values less than *HI* values using *M_O_*_BS_ and *M*_jk_ values drawn from the data.

### 3.2. Evaluation of the Function g in the HI_INT_ Formula

The function *g* is not empirically derived but only heuristically defined. It is used in two ways when applying the *HI*_INT_ formula: (1) to increase an observed binary interaction magnitude, *M*_OBS_, to its theoretical maximum, *M*, based on the binary component fractions; and (2) to reduce *M* from its theoretical maximum based on the component fractions in the assessment mixture, in this case, the quaternary mixture, when the two component exposure levels are not equitoxic, i.e., *HQ*_1_ is not equal to *HQ*_2_. If function *g* is correctly specified, then for two mixtures of different component fractions, *M*^g^ should be the same (for a specific pair and endpoint). Each THM binary mixture dataset included both a 1:1 mixture composition and an environmental composition (Table 3). The evaluation of *g* then involves comparing *M*^g^ across those two mixtures. A sample comparison of these estimated maximum *M* values can be made for the Log(ALT) and ED10 values shown in Table 8 for the 1:1 mixture and in Table 11 for the environmental mixture. The corresponding results for *M* from these two tables, shown in order as (1:1, environmental), are:(1.27, 1.08), (1.04, 1.36), (1.42, 2.18), (1.14, 2.41), (1.16, 5.20), (1.53, 1.46) 
for experiments #2-1 through 2-6, respectively. In this case, the resulting estimates of the theoretical maximum *M* are not very consistent.

## 4. Discussion and Conclusions

### 4.1. Attributes and Improved Understanding of the HI_INT_ Formula

Our objective in this research was to clarify the numerical properties of the *HI*_INT_ formula and to identify any potential problems or weaknesses in the formulaic definition itself, as well as in the implementation of that definition as a risk assessment tool. The example data used are for four well-studied regulated THMs with robust binary and quaternary mixture data. In the initial evaluation of the formula, the numerical behavior with limiting cases was exactly as designed (see Section 1.2). The *HI*_INT_ did agree with the *HI* as the interaction magnitude *M* decreased to zero, and it agreed if the WOE judgment was that the evidence supported dose additivity, resulting in *B* = 0. In many environmental mixture risk assessments, there is one chemical that dominates the exposure and expected toxicity, and *HI*_INT_ does indeed approach the *HQ* for the dominant chemical as the other component fractions are reduced to zero. 

One advantage of the U.S. EPA (2000) *HI*_INT_ formula in Equation (3) over an earlier version [12,13,38] is its ability to reflect the mixture composition, specifically, to reduce the expected interaction magnitude if the component exposures are not equitoxic. That adjustment is by the function *g*. When applying the *HI*_INT_ formula to an actual assessment mixture, the value for each binary *M* used in the *HI*_INT_ formula should be the conceptual maximum calculated in Equation (15). *HQ* values for the assessment mixture that are not equal would then be used in function *g* in Equation (5) to reduce *M* to the expected interaction magnitude for that actual mixture exposure. An important contribution from the research presented here is a clarification of how the binary interaction magnitudes are calculated, how each is converted into a theoretical maximum for use in risk assessment with *HI*_INT_ and how those binary magnitudes are then reduced to reflect a nonequitoxic composition of the assessment mixture. In that explanation, we demonstrated how a 1:1 binary mixture composition may not be equitoxic (i.e., *HQ*_j_ is not equal to *HQ*_k_), even for toxicologically similar chemicals. 

Another advantage of *HI*_INT_ is its relative simplicity as an interaction-based approach. The *HI*_INT_ is an index of concern similar to the *HI*, which has been used successfully for hazardous waste site assessment by the U.S. EPA and State agencies. The formula does require interaction information, but only for binary mixtures, which is likely to be more available than information for higher-complexity mixtures. A decision index need not be very precise or accurate, as long as the information it provides to a risk manager is reproducible and useful. One relevant finding presented here is that a significant interaction magnitude for one pair of components might have a small impact on the HI value. For example (Table 8, based on Env binary mixture), for the Log(ALT) endpoint, the interaction magnitude is 5.2 for CHCl3 with CDBM (10% BMR) but the change in index is small, from HI = 6.03 to HI_INT_ = 4.44 (Table 10). The change in *HI* from the interaction evidence in animal studies would be even less because the U.S. EPA scores for interaction weight of evidence err on the side of public safety (Table 2), in that stronger “less-than-additive” evidence is required before relaxing a dose-additive-based action level or clean up goal than is required with “greater-than-additive” evidence before tightening an action level. 

The estimates of the binary interaction magnitudes *M*_jk_ are better for the 1:1 ratio mixtures because of the larger number of dose groups compared with those for the environmental binary mixtures. Those magnitudes ranged from 1.0 to 1.9 for PcLiv (Table 7) and 1.0 to 1.5 for Log(ALT), somewhat less than the proposed default value of 5 for *M*_jk_ [5], but more accurately reflect what the data show (see example in Table 11). Although the values based on the environmental binary mixtures have more uncertainty, it is interesting that the estimated magnitudes for both sets of binary mixtures were approximately 5 or less, which is consistent with the findings of a survey of the literature on interaction magnitudes at low doses that suggested a range of 3–5-fold [30].

One disadvantage of the *HI*_INT_ in the context of risk characterization and decision-making is that this approach is constrained to one adverse effect, i.e., liver dysfunction in the case of these four THMs. As with the *HI* and with scenario-specific RPF-based assessments, an absence or lack of hazard and/or dose–response data on the specific target organ for a mixture component chemical may result in that component not being adequately accounted for in this approach, potentially underestimating the health risk. That is, an *HI*_INT_ can only be derived for those chemicals for which interaction data are available for the same target organ or system endpoint/effect being assessed. Many chemicals found in the environment have data gaps in epidemiological or animal toxicological dose–response and interaction information for multiple types of health effects. That is why the U.S. EPA recommends the calculation of the *HI* for each health endpoint of concern [9]; the same recommendation should apply to the use of the *HI*_INT_ approach. A screening level *HI*_INT_, similar to its counterpart *HI* that is used for Superfund sites [11], is not recommended because interactions could cross over and influence several health endpoints and so no general conclusion about being health protective is yet justified. 

### 4.2. Possible Improvements

There are some uncertainties with the modeling that affect the accuracy of the results. The function *g* is based on a consensus that the interaction should be maximal when chemicals are at equitoxic levels, although there are too few studies to show this. Our evaluation of function *g* is based in part on comparing the results for the 1:1 binary mixtures with the results for binary mixtures of environmental composition; the latter had only two nonzero dose groups and often had few animals at the highest dose. The dose–response modeling of those mixtures was then highly uncertain, particularly for estimating effective doses at 10% BMR where no data were obtained: for the environmental binary mixtures (Table 5b), *D*_j_10 < 0.7 mmol/kg/day for both endpoints in all studies, whereas the nonzero doses tested were 1.0 and 3.0 mmol/kg/day. To better investigate the assumption that the interaction should be maximal when chemicals are at equitoxic levels, one study should have component ratios set to give equitoxic fractions (e.g., *HQ*_1_ = *HQ*_2_) with a parallel study with fairly unequal *HQ*s, using at least four nonzero doses to help define the dose–response curvature. Without such data, the evaluation of the accuracy of the function *M*^g^ for interaction magnitude is incomplete. That function, not just the exponent *g*, should be replaced by an empirically based function once further binary results become available for a variety of component fractions. 

The most complex mixture evaluated here only involved four chemicals, and ones that are fairly similar in toxic potency (D10 mean values in Table 4 with a range of 0.33−0.53 mmol/kg/day). It would be interesting to carry out a similar evaluation first for a mixture of twice as many chemicals, even ones with similar potency, to see if the conservatism of this index increases with the number of components. There is evidence that some groups of fairly similar chemicals have relative potencies that change with the response magnitude [9,39]. It would then be enlightening to conduct this evaluation with chemicals that have not only more variation in relative potency, but also relative potencies that change with response level. 

Evaluation of the *HI*_INT_ as applied to human health risk assessment is problematic. The binary weight of evidence score (*B*) decreases the impact of the interaction as the evidence becomes weaker. That characteristic of the procedure is a science policy decision, and not yet amenable to statistical evaluation. The ATSDR has a WOE scheme that breaks down the evidence into multiple categories, including whether there is direct evidence of impact on toxic effects and whether the data are for the correct exposure route. The U.S. EPA approach is much simpler, using a judgment of the extent of extrapolation required (Table 2). The U.S. EPA scoring is biased in the direction of safety. That bias further complicates the ability to evaluate the accuracy of the *HI*_INT_ formula. In spite of those uncertainties, the results in this small study are encouraging. When there are no extrapolation penalties (so *B* = 1 or *B* = −1), the *HI*_INT_ moves from the *HI* in the direction indicated by the binary interactions, greater concern when based on PcLiv and less concern when based on Log(ALT) (Table 10). Furthermore, the accuracy of the predicted mixture response is improved compared with that using dose addition: the estimates of effective dose based on *HI*_INT_ are generally closer to the actual mixture effective dose (Table 9). These comparisons with the *HI* indicate the consequences of using a default regulatory formula that does not include interaction results. But not to be ignored is the most important weakness in the *HI*_INT_ formula, namely, the reliance on interaction data; even though only binary mixtures data are required, such data are rare. At this point in time, only the ATSDR has an active program in the federal sector for evaluating binary toxicologic interactions for use in health risk assessment, and only a few binary interactions have been evaluated to date [12,40]. Even fewer are the binary studies that have quantified the magnitude of the interaction. New approach methodologies (NAMs) have potential for estimating these missing pieces of information, from grouping by toxicological similarity to kinetic and dynamic interactions. For successful adoption in quantitative mixture risk assessment, there is a need for a structure for accepting and using such information [9,41] and a need for approaches for quantifying the uncertainty in those uses of NAM results [42,43]. 

In summary, the evaluation presented here demonstrates how the *HI*_INT_ formula can be quantitatively evaluated in terms of dose–response modeling. When interpreted as a dose–response formula, these preliminary results suggest that this formula may, under some circumstances, be more accurate than dose addition and more accurate than the default assumption of interaction magnitude = 5. The possible over-interpretation of its accuracy regarding interactions is lessened by the WOE scoring, which adjusts for the quality of interaction evidence. This formula allows information on toxic interaction to quantitatively affect the risk assessment. It is expected to be useful, at least for simple mixtures of well-studied chemicals with interaction information, in determining a quantitative indicator of health risk to aid decision-making. 

## Figures and Tables

**Figure 1 toxics-12-00305-f001:**
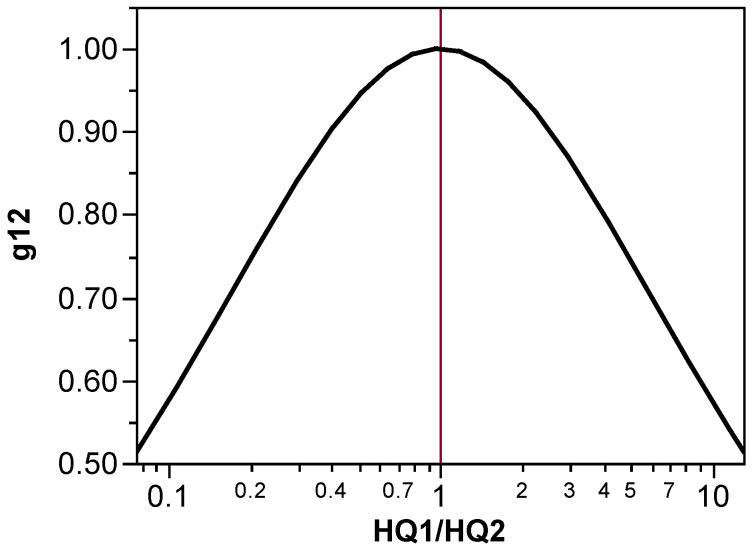
The function *g* used in the interaction-based hazard index formula, showing a maximum when the two chemicals are roughly equitoxic (equal hazard quotients).

**Figure 2 toxics-12-00305-f002:**
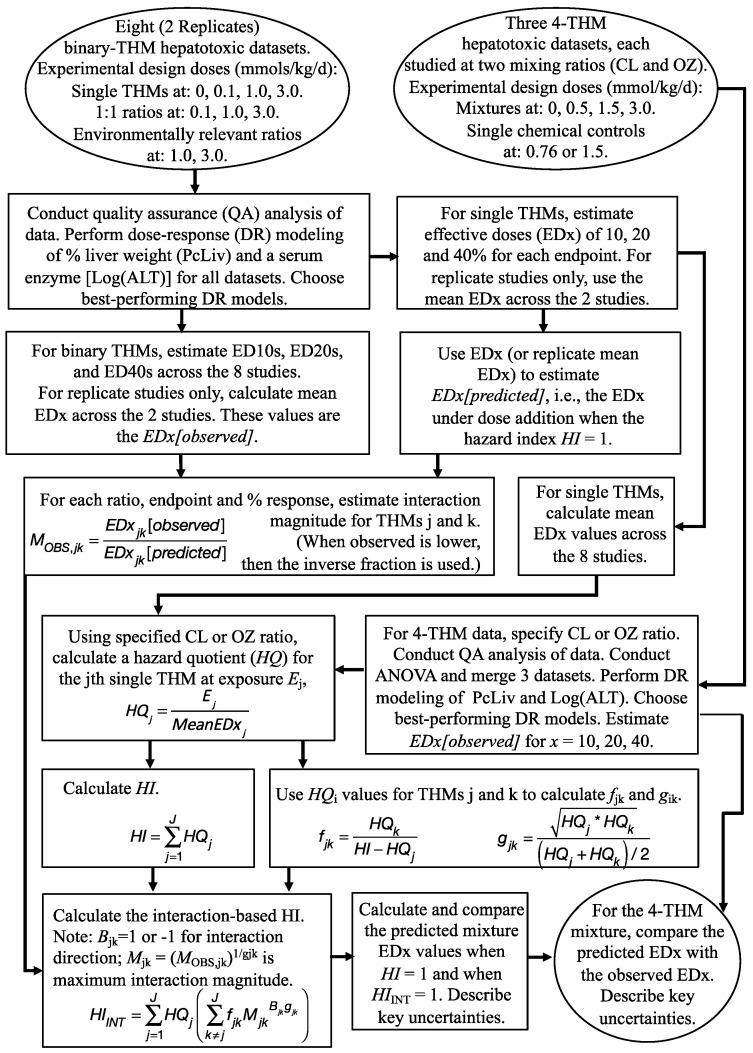
Calculation steps for evaluating the interaction-based hazard index. In binary studies, single THMs are only in 3 to 5 studies each. Control groups are in every study. Note that to calculate the maximum interaction magnitude [*M*_jk_ = (*M*_OBS,jk_)^1/gjk^], the *g* function is based on the component fractions in the binary mixture where *M*_OBS_ was determined; to calculate the *HI*_INT_, the *g* function is based on the component fractions in the quaternary mixture.

**Figure 3 toxics-12-00305-f003:**
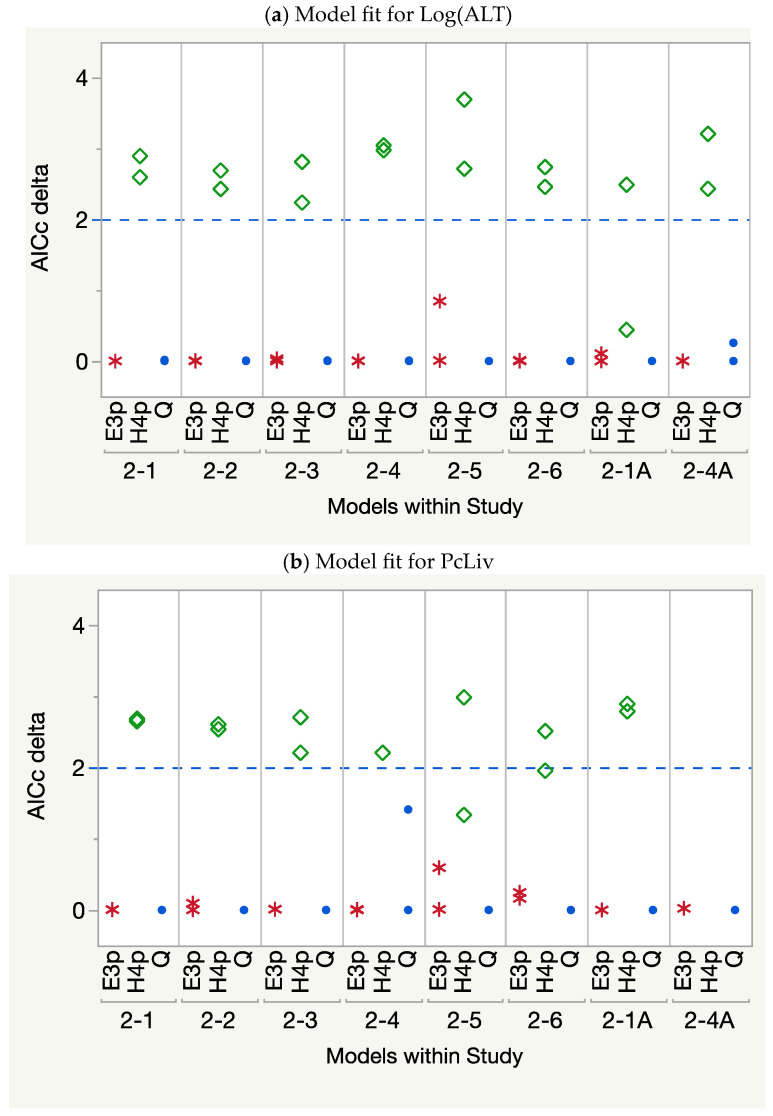
Comparison of fits of three models to binary single data across studies by *AICc delta* (change in *AICc* from the minimum across the three models). The 3-parameter exponential (E3p, denoted Exp-3P in the text) is consistently the best model or indistinguishable from best. *AICc delt*a = 0 indicates the model with minimum *AICc*. When two points are shown for a model, they represent different fits for the two component chemicals (e.g., E3p model in study 2-5). If only one point shows, both chemicals had the same *AICc* for that model. With study 2-4A, the H4p model had *AICc* more than 5 units higher than the minimum *AICc*. Models: (star) E3p = 3-parameter exponential; (rhombus) H4p = 4-parameter Hill; (dot) Q = quadratic.

**Figure 4 toxics-12-00305-f004:**
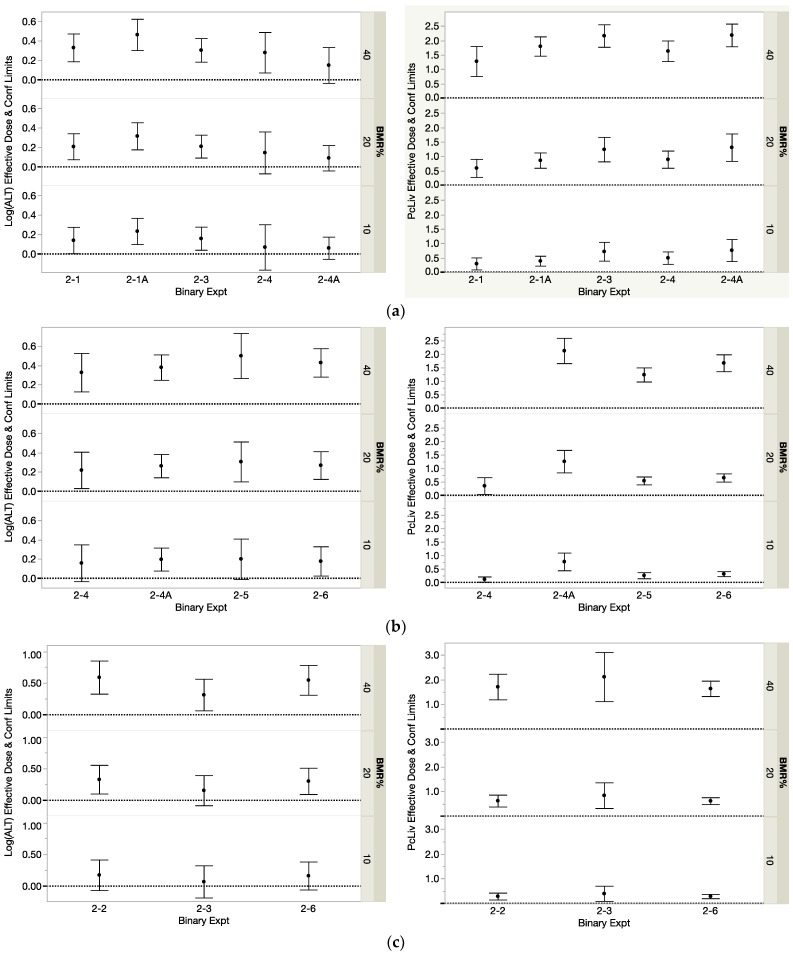
(**a**). Effective dose (*D*j10, *D*j20, *D*j40) estimates with 95% confidence limits for Log(ALT) (**left**) and PcLiv (**right**) for BDCM across the binary studies. (**b**). Effective dose (*D*_j_10, *D*_j_20, *D*_j_40) estimates with 95% confidence limits for Log(ALT) (**left**) and PcLiv (**right**) for CDBM across the binary studies. Results for *D*_j_10 and *D*_j_20 for PcLiv in study 2-4A are statistically different from corresponding values in the other three studies. No *D*_j_40 results for PcLiv are plotted for study 2-4 because only one animal survived at the high dose level (3.0 mmol/kg/day), precluding effective dose estimates at high response levels. (**c**). Effective dose (*D*_j_10, *D*_j_20, *D*_j_40) estimates with 95% confidence limits for Log(ALT) (**left**) and PcLiv (**right**) for CHBr3 across the binary studies. (**d**). Effective dose (*D*_j_10, *D*_j_20, *D*_j_40) estimates with 95% confidence limits for Log(ALT) (**left**) and PcLiv (**right**) for CHCl3 across the binary studies.

**Figure 5 toxics-12-00305-f005:**
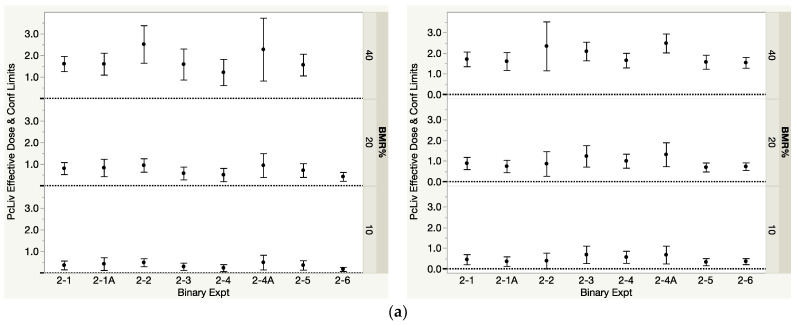
(**a**). Effective dose (*D*_MIX2_) estimates for 10, 20, 40% response levels with 95% confidence limits for PcLiv across the 1:1 (**left**) and environmental ratio (**right**) binary mixtures. Having 4 dose groups in the 1:1 mixture vs. 3 in the environmental mixture only reduced the effective dose variance in some studies. No *D*_MIX2_40 results are plotted for study 2-6, 1:1 mixture because that response level was at the upper asymptote, so the *D*_MIX2_ confidence interval was large and highly uncertain (−16.2, 22.5). (**b**). Effective dose (*D*_MIX2_) estimates for 10, 20, 40% response levels with 95% confidence limits for Log(ALT) across the 1:1 (**left**) and environmental ratio (**right**) binary mixtures. Having 4 dose groups in the 1:1 mixture vs. 3 in the environmental mixture only reduced the effective dose variance in some studies.

**Figure 6 toxics-12-00305-f006:**
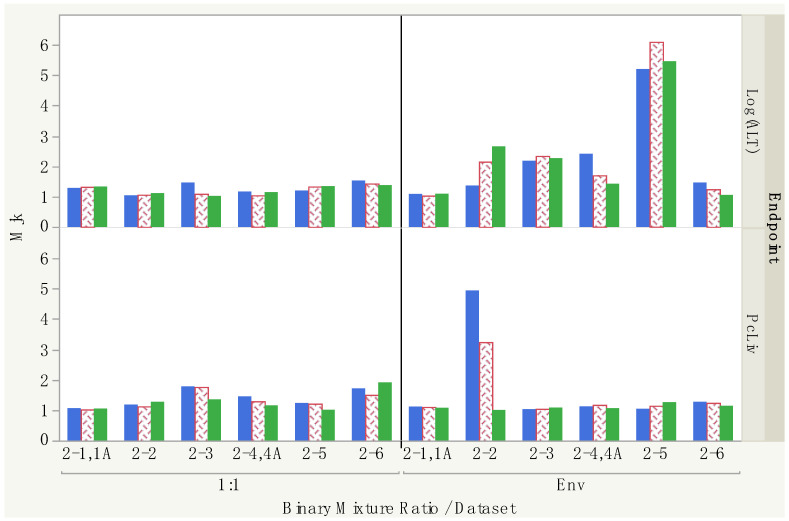
Interaction magnitude (*M*_jk_) values for PcLiv and Log(ALT) for the 1:1 and environmental mixtures across three BMR levels (10%, 20%, 40% in bar groups, left to right). Similar bar heights show consistency of *M*_jk_ values for most BMR level and endpoint combinations. Experiment #2-2, environmental composition, for PcLiv of CHCl3 with CHBr3 shows the largest differences in *M*_jk_ values across the BMR levels.

**Table 1 toxics-12-00305-t001:** Notation for interaction-based hazard index analyses.

Notation	Description
BMRx	Benchmark response value for PcLiv or for ALT, calculated as x% increase from the control mean value of that endpoint and used with inverse prediction to obtain the effective dose (EDx) for that endpoint; for ALT, the inverse prediction used Log(BMR)
D_j_	EDx estimated by modeling the jth component
D_MIX2,_ D_MIX4_	EDx estimated by modeling the binary (MIX2) or quaternary (MIX4) mixture data
D_ADD2_, D_ADD4_	EDx calculated for the mixture using dose addition and single chemical data for the binary (ADD2) or quaternary (ADD4) mixtures
D_INT4_	EDx calculated for the mixture using the EPA interaction formula
π_j_	Dose fraction in the mixture for the jth component
HQ_j_	Hazard quotient for the jth component, in this research, using animal data
HI	Additive hazard index
HI_INT_	Interaction-based HI
M_OBS_	Observed interaction magnitude calculated from binary data
B_jk_	Binary weight of evidence score for the strength of evidence that chemical k will influence the toxicity of chemical j in humans; the sign of B indicates the direction of the interaction
f_jk_	Toxic hazard of the kth chemical relative to the total hazard from all chemicals potentially interacting with chemical j
g_jk_	Degree to which chemicals j and k are present in equitoxic amounts in the mixture; the HQ_j_ values used to calculate g_jk_ depend on the component fractions of either the binary or quaternary mixture along with the component effective doses
M_jk_	Theoretical maximum interaction magnitude

**Table 2 toxics-12-00305-t002:** Weight-of-evidence scoring (*B*_jk_) used by US EPA (2000) for binary interaction data.

Category	Description	Greater Than Add	Less Than Add
I	Directly relevant to humans	1.0	−1.0
II	Animal studies, but relevant	0.75	−0.5
III	Plausible evidence, possibly not relevant because of different exposure route or use of in vitro data	0.5	0.0
IV	Additivity demonstrated or assumed because of poor data	0.0	0.0

**Table 3 toxics-12-00305-t003:** Description of the experiments.

Exp. ID ^a^	Components Involved	Ratios Studied		Dosages Studied mmol/kg/day ^b^
		1:1	Environmental	
#2-1, #2-1A	CHCl3:BDCM	1:1	2.7:1	Single THMs individually at: 0.1, 1.0, 3.0 1:1 ratios mixtures at: 0.1, 1.0, 3.0 Environmentally relevant ratios mixtures at: 1.0, 3.0
#2-2	CHCl3:CHBr3	1:1	65:1	
#2-3	BDCM:CHBr3	1:1	24:1	
#2-4, #2-4A	BDCM:CDBM	1:1	2.4:1	
#2-5	CHCl3:CDBM	1:1	6.5:1	
#2-6	CDBM:CHBr3	1:1	10:1	
#4-2	CHCl3:BDCM :CDBM:CHBr3	Chlorination Ratio 0.319:0.342:0.290:0.049 Ozonation Ratio 0.655:0.187:0.129:0.029 (ratios apply to all three quaternary mixtures)		Each quaternary mixture studied at: 0.05, 1.5, 3.0 Each quaternary THM studied at one level: 0.76 (CDBM), 1.5 (other THMs)
#4-3	CHCl3:BDCM :CDBM:CHBr3			
#4-4	CHCl3:BDCM :CDBM:CHBr3			

^a^ Experiments 2-1 and 2-1A are replicates; 2-4 and 2-4A are replicates. These ID numbers represent the number of chemicals in the experiment–study number and are used in the other tables and figures; this configuration will also be used in other papers from this project. ^b^ Beginning dose group sizes were 8–20 female CD-1 mice. Not all animals survived. Each study included a control group (zero dose).

**Table 4 toxics-12-00305-t004:** Single-THM *D*_j_ values (mmol/kg/day) for PcLiv and Log(ALT) with replicate experiment values averaged based on the binary studies ^a^.

THM Modeled	Studied with	Binary Exp	PcLiv			Log(ALT)		
			Dj10	Dj20	Dj40	Dj10	Dj20	Dj40
BDCM	CHCl3	2-1	0. 30	0.59	1.28	0.14	0.21	0.33
BDCM	CHCl3	2-1A	0.39	0.86	1.80	0.23	0.32	0.46
2-1, 2-1A Average			0.34	0.73	1.54	0.19	0.26	0.40
BDCM	CHBr3	2-3	0.72	1.24	2.16	0.16	0.21	0.30
BDCM	CDBM	2-4	0.50	0.89	1.63	0.07	0.14	0.28
BDCM	CDBM	2-4A	0.76	1.31	2.19	0.06	0.09	0.15
2-4, 2-4A Average			0.63	1.10	1.91	0.06	0.12	0.21
BDCM Mean Dj			0.53	0.98	1.81	0.13	0.19	0.31
CDBM	BDCM	2-4	0.12	0.35	-	0.16	0.22	0.33
CDBM	BDCM	2-4A	0.77	1.26	2.13	0.20	0.26	0.38
2-4, 2-4A Average			0.44	0.81	2.13	0.18	1.24	0.35
CDBM	CHCl3	2-5	0.26	0.54	1.24	0.20	0.31	0.50
CDBM	CHBr3	2-6	0.32	0.65	1.67	0.18	0.27	0.43
CDBM Mean Dj			0.36	0.70	1.68	0.18	0.26	0.41
CHCl3	BDCM	2-1	0.64	1.18	2.06	0.29	0.46	0.73
CHCl3	BDCM	2-1A	0.38	0.79	1.70	0.31	0.45	0.68
2-1, 2-1A Average			0.51	0.98	1.88	0.30	0.46	0.71
CHCl3	CHBr3	2-2	0.73	1.33	2.36	0.20	0.30	0.49
CHCl3	CDBM	2-5	0.33	0.65	2.06	0.06	0.09	0.17
CHCl3 Mean Dj			0.52	0.99	2.05	0.21	0.33	0.52
CHBr3	CHCl3	2-2	0.29	0.62	1.71	0.18	0.33	0.59
CHBr3	BDCM	2-3	0.40	0.84	2.12	0.07	0.16	0.32
CHBr3	CDBM	2-6	0.29	0.62	1.64	0.16	0.30	0.55
CHBr3 Mean Dj			0.33	0.70	1.83	0.14	0.26	0.49

^a^ The component “means” are across all studies that included that component. See Figure 4 parts a–d for 95% confidence intervals around the means. Component “averages” are across the two replicate studies: 2-1 with 2-1A, 2-4 with 2-4A.

**Table 5 toxics-12-00305-t005:** Binary THM *D*_MIX2_ values (mmol/kg/day) for PcLiv and Log(ALT) with replicate experiment values averaged, for (a) 1:1 ratio and (b) environmental ratio ^a^.

(**a**)							
**THMs in Binary Mixture**	**Binary** **Exp**	**PcLiv**			**Log(ALT)**		
		**D_MIX2_10**	**D_MIX2_20**	**D_MIX2_10**	**D_MIX2_20**	**D_MIX2_10**	**D_MIX2_20**
BDCM, CHCl3	2-1	0.36	0.80	1.61	0.30	0.46	0.71
BDCM, CHCl3	2-1A	0.42	0.83	1.60	0.29	0.42	0.63
2-1, 2-1A Average		0.39	0.82	1.60	0.29	0.44	0.67
CHCl3, CHBr3	2-2	0.48	0.95	2.51	0.19	0.34	0.59
BDCM, CHBr3	2-3	0.30	0.58	1.59	0.14	0.20	0.30
BDCM, CDBM	2-4	0.23	0.51	1.21	0.12	0.17	0.26
BDCM, CDBM	2-4A	0.49	0.94	2.27	0.09	0.13	0.21
2-4, 2-4A Average		0.36	0.73	1.74	0.11	0.15	0.24
CDBM, CHCl3	2-5	0.36	0.71	1.56	0.10	0.18	0.33
CDBM, CHBr3	2-6	0.18	0.42	3.16	0.26	0.41	0.66
(**b**)							
**THMs in Binary Mixture**	**Binary** **Exp**	**PcLiv**			**Log(ALT)**		
		**D_MIX2_10**	**D_MIX2_20**	**D_MIX2_40**	**D_MIX2_10**	**D_MIX2_20**	**D_MIX2_40**
BDCM, CHCl3	2-1	0.46	0.89	1.71	0.17	0.27	0.44
BDCM, CHCl3	2-1A	0.36	0.75	1.61	0.31	0.51	0.82
2-1, 2-1A Average		0.41	0.82	1.66	0.24	0.39	0.63
CHCl3, CHBr3	2-2	0.39	0.87	2.35	0.21	0.36	0.60
BDCM, CHBr3	2-3	0.69	1.24	2.10	0.10	0.14	0.22
BDCM, CDBM	2-4	0.57	1.00	1.65	0.11	0.17	0.26
BDCM, CDBM	2-4A	0.68	1.32	2.49	0.17	0.24	0.38
2-4, 2-4A Average		0.62	1.16	2.07	0.14	0.21	0.32
CDBM, CHCl3	2-5	0.33	0.70	1.58	0.12	0.22	0.39
CDBM, CHBr3	2-6	0.36	0.74	1.55	0.14	0.24	0.43

^a^ See Figure 5 parts a, b for 95% confidence intervals around the means.

**Table 6 toxics-12-00305-t006:** Details of calculation of *M*_jk_ from single-THM and 1:1 binary mixture data based on *D*_j_10 for PcLiv.

Dataset ^a^ and Type of Dose		Observed 10% Dose	Predicted ^b^ (D_ADD2_10)	M_jk_ ^c^	D_ADD2_-D_MIX2_	Direction ^d^
CHCl3	2-1,1A *D*_j_	0.51				
1:1	2-1,1A *D*_MIX2_	0.39	0.41	1.06	0.02	additive
BDCM	2-1,1A *D*_j_	0.34				
CHCl3	2-2 *D*_j_	0.73				
1:1	2-2 *D*_MIX2_	0.48	0.42	1.18	−0.06	<additive
CHBr3	2-2 *D*_j_	0.29				
BDCM	2-3 *D*_j_	0.72				
1:1	2-3 *D*_MIX2_	0.30	0.51	1.78	0.21	>additive
CHBr3	2-3 *D*_j_	0.40				
BDCM	2-4,4A *D*_j_	0.63				
1:1	2-4,4A *D*_MIX2_	0.36	0.52	1.45	0.16	>additive
CDBM	2-4,4A *D*_j_	0.44				
CHCl3	2-5 *D*_j_	0.33				
1:1	2-5 *D*_MIX2_	0.36	0.29	1.23	−0.07	<additive
CDBM	2-5 *D*_j_	0.26				
CDBM	2-6 *D*_j_	0.32				
1:1	2-6 *D*_MIX2_	0.18	0.30	1.71	0.12	>additive
CHBr3	2-6 *D*_j_	0.29				

^a^ The dataset identification is the binary experiment (Table 3) and the single component or mixture dataset in that experiment. Note that in this table, the effective dose within the same binary experiment (including replicate averages) is used for the interaction magnitude calculation, whereas the mean effective dose for each THM in Table 4 is used for the *HQ* calculations. The binary mixture mean effective dose is from Table 5(a). ^b^ Predicted *D*_ADD2_ under dose addition for *HI* = 1, 1:1 mixture. ^c^ Theoretical maximum interaction magnitude calculated as *M*_jk_ = *M*_OBS_^1/g^ using the component effective doses and proportions in the binary mixtures to determine the value of *g*. ^d^ Additive here refers to dose addition. In this research, *M*_jk_ < 1.10 is considered dose-additive. When <additive, then *D*_ADD2_-*D*_MIX2_ is negative, so *B* = −1 in the *HI*_INT_ formula (similarly, for >additive, then *B* = 1).

**Table 7 toxics-12-00305-t007:** Calculated binary mixture interaction magnitudes (*M*_jk_) for PcLiv ^a^.

Dataset			M_jk_ (10%)	Direction ^b^ (10%)	M_jk_ (20%)	Direction ^b^ (20%)	M_jk_ (40%)	Direction ^b^ (40%)
CHCl3:BDCM	2-1,1A	1:1	1.06		1.02		1.05	
CHCl3:CHBr3	2-2	1:1	1.18	<add	1.12	<add	1.27	<add
BDCM:CHBr3	2-3	1:1	1.78	>add	1.76	>add	1.35	>add
BDCM:CDBM	2-4,4A	1:1	1.45	>add	1.29	>add	1.15	>add
CHCl3:CDBM	2-5	1:1	1.23	<add	1.21	<add	1.01	
CDBM:CHBr3	2-6	1:1	1.71	>add	1.50	>add	1.91	<add
CHCl3:BDCM	2-1,1A	Env	1.11	>add	1.10	>add	1.07	
CHCl3:CHBr3	2-2	Env	4.94	>add	3.24	>add	1.0004	
BDCM:CHBr3	2-3	Env	1.03		1.04		1.08	
BDCM:CDBM	2-4,4A	Env	1.12	<add	1.17	<add	1.06	
CHCl3:CDBM	2-5	Env	1.04		1.14	<add	1.26	>add
CDBM:CHBr3	2-6	Env	1.27	<add	1.24	<add	1.14	>add

^a^ Theoretical maximum interaction magnitudes calculated as *M*_jk_ = *M*_OBS_^1/g^ using the component effective doses and proportions in the binary mixtures to determine the value of *g*. ^b^ An *M*_jk_ < 1.10 indicates a dose-additive effect, and this designation is removed from the Direction column to enhance visual information. <add and >add refer to less than and greater than dose-additive, respectively.

**Table 8 toxics-12-00305-t008:** Calculated binary mixture interaction magnitudes (*M*_jk_) for Log(ALT) ^a^.

Dataset			M_jk_ (10%)	Direction ^b^ (10%)	M_jk_ (20%)	Direction ^b^ (20%)	M_jk_ (40%)	Direction ^b^ (40%)
CHCl3:BDCM	2-1,1A	1:1	1.28	<add	1.32	<add	1.33	<add
CHCl3:CHBr3	2-2	1:1	1.04		1.06		1.11	<add
BDCM:CHBr3	2-3	1:1	1.46	<add	1.09		1.02	
BDCM:CDBM	2-4,4A	1:1	1.16	<add	1.04		1.14	>add
CHCl3:CDBM	2-5	1:1	1.20	<add	1.33	<add	1.34	<add
CDBM:CHBr3	2-6	1:1	1.53	<add	1.43	<add	1.37	<add
CHCl3:BDCM	2-1,1A	Env	1.08		1.03		1.09	
CHCl3:CHBr3	2-2	Env	1.36	<add	2.15	<add	2.65	<add
BDCM:CHBr3	2-3	Env	2.18	>add	2.34	>add	2.26	>add
BDCM:CDBM	2-4,4A	Env	2.41	<add	1.70	<add	1.42	<add
CHCl3:CDBM	2-5	Env	5.20	<add	6.10	<add	5.46	<add
CDBM:CHBr3	2-6	Env	1.46	>add	1.24	>add	1.05	

^a^ Theoretical maximum interaction magnitudes calculated as *M*_jk_ = *M*_OBS_^1/g^ using the component effective doses and proportions in the binary mixtures to determine the value of *g*. ^b^ An *M*_jk_ < 1.10 indicates a dose-additive effect, and this designation is removed from the Direction column to enhance visual information. <add and >add refer to less than and greater than dose-additive, respectively.

**Table 9 toxics-12-00305-t009:** Quaternary mixture predicted effective dose results (mmol/kg/day) for PcLiv and Log(ALT).

Predicted Dose	Chlorination						Ozonation					
	PcLiv			Log(ALT)			PcLiv			Log(ALT)		
	10%	20%	40%	10%	20%	40%	10%	20%	40%	10%	20%	40%
D_MIX4_ ^a^	0.355	0.722	1.679	0.110	0.240	0.473	0.284	0.649	1.629	0.288	0.522	0.894
D_ADD4_ ^b^	0.455	0.865	1.837	0.166	0.249	0.393	0.492	0.926	1.938	0.185	0.281	0.444
D_INT4_ (1:1) ^c^	0.396	0.797	1.741	0.197	0.279	0.427	0.479	0.937	1.892	0.220	0.339	0.540
D_INT4_ (Env) ^c^	0.417	0.872	1.707	0.225	0.345	0.537	0.435	0.887	1.764	0.226	0.377	0.612

^a^ Estimated from inverse prediction using the modeled combined quaternary mixture datasets. ^b^
*D*_ADD_ and *D*_INT_ values used *D*_j_ calculated using component “means” in Table 4. ^c^ *D*_INT4_ values were calculated using the theoretical maximum interaction magnitude, *M*_jk_, from each binary mixture and values of *g* determined using the component proportions of the quaternary mixtures.

**Table 10 toxics-12-00305-t010:** Quaternary mixture *HI* and *HI*_INT_ results for PcLiv and Log(ALT) for mixture total dose = 1.0 mmol/kg/day.

Predicted Index	Chlorination						Ozonation					
	PcLiv			Log(ALT)			PcLiv			Log(ALT)		
	10%	20%	40%	10%	20%	40%	10%	20%	40%	10%	20%	40%
HI	2.197	1.156	0.544	6.033	4.022	2.547	2.032	1.080	0.516	5.398	3.564	2.255
HI_INT_ (1:1) ^a^	2.524	1.254	0.574	5.067	3.579	2.344	2.086	1.067	0.529	4.546	2.988	1.851
HI_INT_ (Env) ^a^	2.396	1.147	0.586	4.440	2.895	1.863	2.299	1.128	0.567	4.420	2.653	1.634

^a^ *HI*_INT_ values were calculated using the theoretical maximum interaction magnitude, *M*_jk_, from each binary mixture and values of *g* determined using the component proportions of the quaternary mixtures.

**Table 11 toxics-12-00305-t011:** Demonstration of *M*_OBS_ vs. *M*_jk_ results using 10% *D*_j_ and environmental mixture *D*_MIX2_ for Log(ALT).

Binary Datasets		M_OBS,jk_	g_jk_	M_jk_	M_jk_ Default	Quaternary Chlorination Results	
						Indices ^a^	Doses ^b^
CHCl3:BDCM	2-1, 2-1A	1.08	0.97	1.08	5	HI = 6.033 HI_INT_ = 4.921 using M_OBS,jk_ HI_INT_ = 4.440 using M_jk_ HI_INT_ = 12.091 using default of M_jk_ = 5	0.166 for HI = 1 0.203 for HI_INT_ = 1 using M_OBS,jk_ 0.225 for HI_INT_ = 1 using M_jk_ 0.083 for HI_INT_ = 1 using default of M_jk_ = 5
CHCl3:CHBr3	2-2	1.08	0.26	1.36	5		
BDCM:CHBr3	2-3	1.54	0.56	2.18	5		
BDCM:CDBM	2-4, 2-4A	1.81	0.67	2.41	5		
CHCl3:CDBM	2-5	1.93	0.40	5.20	5		
CDBM:CHBr3	2-6	1.25	0.59	1.46	5		

^a^ Index calculated for total mixture dose = 1.0 mmol/kg/day. ^b^ Units of mmol/kg/day.

## Data Availability

Data files for this project can be found at U.S. EPA’s ScienceHub website. Online, https://catalog.data.gov/dataset/epa-sciencehub (accessed on 16 April 2024).

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
