# Peer review of "Evaluation of the Interaction-Based Hazard Index Formula Using Data on Four Trihalomethanes from U.S. EPA’s Multiple-Purpose Design Study"

_toxics, 2024, doi:10.3390/toxics12050305_

Round 1

Reviewer 1 Report

Comments and Suggestions for Authors

 General comments:

The authors of the manuscript present a sound case study on the interaction-based hazard index and demonstrate how the HIINT formula can be quantitatively evaluated in terms of dose-response modelling. The study is based on four trihalomethanes and is presented in a comprehensive and clear manner. The analysis are sound and results are well presented. I appreciate the proposed improvements of the method.

However, I would prefer to restructure the introduction (see specific comments), it increases comprehensibility in my opinion.

Specific comments:

Page1, line 20: Introduce abbreviation ALT

Page 1, lines 33-34: The European laws require to take effects of mixtures into account, however, the do not refer to specific methods! Methods are usually proposed in guidelines of ECHA or EFSA. Better:

“In addition, several laws in Europe require to take cumulative and synergistic effects into account (Kienzler et al., 2016;https://doi.org/10.1016/j.yrtph.2016.05.020), furthermore several approaches and specific methodologies have been proposed to address mixtures [1; Bopp et al., 2019; https://doi.org/10.1080/10408444.2019.1579169).”

The additional publications give a good overview on legal requirements and methodologies within the EU.

Page 2, line 58: delete second point

Page 2, lines 61 -80: I propose to shift the section (line 61 to 80) to line 49, just before the sentence “The four... “. Furthermore I suggest to shift the sentence “The four trihalomethanes (THMs) that were chosen have the 49 liver as a sensitive target organ in common, thus satisfying the similarity property [11].” To line 58. (insert after “quaternary mixtures.“)

Page 3, line 101: I suggest to change from “is motivated by” to “is based on the assumption of”

Page 4. Line 120: As an example, the European Chemicals Agency (ECHA) proposed a tiered approach for practical risk assessment of biocidal products, which is based on refinement steps for calculated HQs/HIs. (ECHA, 2017).

Ref: Guidance on the Biocidal Products Regulation; Volume III Human Health - Assessment & Evaluation, (Parts B+C), Version 4.0, December 2017, Page 268 (https://echa.europa.eu/documents/10162/23036412/biocides_guidance_human_health_ra_iii_part_bc_en.pdf/30d53d7d-9723-7db4-357a-ca68739f5094)

Page 9, Table 3: Please explain where the environmental ratios come from.

Page 16, Figure 3: Please, introduce also the symbols in the caption.

Page 18, Figure 4b, line 587: delete second point.

Reviewer 2 Report

Comments and Suggestions for Authors

General comments:

In the manuscript entitled “Evaluation of the interaction-based hazard index formula using data on four trihalomethanes from U.S. EPA’s multiple-purpose design study”, the authors evaluated the interaction-based hazard index (HIint) approach for the mixture risk assessment of four different trihalomethanes (THMs) on the hepatotoxicity. Since practical concepts and strategies important to implement more reliable mixture risk assessment are still lacking, this study seems to be highly impressive to different sectors including human health, the environment, and the chemical industry. This manuscript is well organized with in-depth supporting references. I therefore recommend acceptance of this manuscript after the minor revision for the publication in Toxics.

Specific comments:

- Lines 57 and 90 on page 2: The repetition of both the full name and its corresponding abbreviation explanation (e.g., THMs), both of which have already been abbreviated within the text, is noted.

 - Abstract and Table 1: The abbreviations, PcLiv and ALT, need to be explained.

- Reference: It is advisable to recheck the table and citation format of the reference to ensure its adherence to the guidelines outlined by the journal.
